# Tumor derived UBR5 promotes ovarian cancer growth and metastasis through inducing immunosuppressive macrophages

Mei Song[1], Oladapo O. Yeku[2,3], Sarwish Rafiq [2,4], Terence Purdon[2], Xue Dong[1], Lijing Zhu[5], Tuo Zhang [1], Huan Wang[6], Ziqi Yu[1], Junhua Mai [7], Haifa Shen[7], Briana Nixon[8], Ming Li [8], Renier J. Brentjens [2] & Xiaojing Ma [1,6✉]

Immunosuppressive tumor microenvironment (TME) and ascites-derived spheroids in ovarian cancer (OC) facilitate tumor growth and progression, and also pose major obstacles for cancer therapy. The molecular pathways involved in the OC-TME interactions, how the crosstalk impinges on OC aggression and chemoresistance are not well-characterized. Here, we demonstrate that tumor-derived UBR5, an E3 ligase overexpressed in human OC associated with poor prognosis, is essential for OC progression principally by promoting tumor-associated macrophage recruitment and activation via key chemokines and cytokines. UBR5 is also required to sustain cell-intrinsic β-catenin-mediated signaling to promote cellular adhesion/colonization and organoid formation by controlling the p53 protein level. OC-specific targeting of UBR5 strongly augments the survival benefit of conventional chemotherapy and immunotherapies. This work provides mechanistic insights into the novel oncogene-like functions of UBR5 in regulating the OC-TME crosstalk and suggests that UBR5 is a potential therapeutic target in OC treatment for modulating the TME and cancer stemness.

[1] Department of Microbiology and Immunology, Weill Cornell Medicine, 1300 York Avenue, New York, NY 10065, USA. [2] Department of Medicine, Memorial Sloan Kettering Cancer Center, New York, NY 10065, USA. [3] Gynecologic Cancers Program, Massachusetts General Hospital, Harvard Medical School, Boston, MA 02114, USA. [4] Department of Hematology and Medical Oncology, Winship Cancer Institute of Emory University School of Medicine, 1760 Haygood Drive, Atlanta, GA 30322, USA. [5] The Comprehensive Cancer Centre of Drum Tower Hospital, Medical School of Nanjing University, 210008 Nanjing, China. [6] Sheng Yushou Center of Cell Biology and Immunology, School of Life Science and Biotechnology, Shanghai Jiao Tong University, 200240 Shanghai, China. [7] Department of Nanomedicine, Houston Methodist Research Institute, Houston, TX 77030, USA. [8] Immunology Program, Memorial Sloan-Kettering Cancer Center, New York, NY 10065, USA. ✉email: xim2002@med.cornell.edu

Ovarian cancer (OC) is the most fatal gynecological malignancy worldwide. Over 22,000 American women are diagnosed with OC each year and approximately 14,000 die of this disease, rendering it the fifth leading cause of cancer deaths among women in the United States[1]. Only about 45% of women with OC survive for five years or longer since diagnosis[2]. Despite the gold standard of debulking surgery and platinum/taxane-based chemotherapy resulting in initial responses in approximately 75% of patients, most women experience relapses and succumb with chemoresistant disease[3]. Therefore, strategies for second-line or alternative therapies are strongly called for.

Unlike many epithelial cancers that spread predominantly through the vasculature, OC has a tendency to disseminate by dislodging from the primary tumor into the peritoneal cavity and implanting on abdominal organs[4]. During this process, the disseminated cells aggregate and form spheroid-like structures to overcome anoikis, immune attack, and chemotherapy[5,6]. As such, spheroid formation is a key step in OC metastatic spread and contributes to recurrence.

The tumor microenvironment (TME) of OC is known to be highly immunosuppressive, which allows evasion of immune surveillance and unhampered tumor growth[7]. The inhibitory cellular microenvironment has been reported to consist of tumor-associated macrophages (TAMs), regulatory T cells (Tregs), and myeloid-derived suppressor cells (MDSCs) as well as tumor-associated dendritic cells (tDCs). TAMs represent a preponderant infiltrating immune population in human OC[8]. TAMs have an immunosuppressive M2 phenotype in TME of OC, which is associated with their ability to promote cancer growth, invasion, angiogenesis, immune evasion, and metastasis[9]. The higher frequencies of M2 subsets (CD163$^+$ TAMs) from malignancy-associated ascites strongly correlated with higher tumor grade and worse progression-free survival[10,11]. Thus, strategies aiming to target vital molecules regulating OC–TAM interaction and spheroid formation may provide opportunities for more efficacious treatment of OC.

UBR5 (Ubiquitin protein ligase E3 component n-recognin 5, also known as EDD) is a HECT (homologous to E6AP C-terminus) domain-containing ubiquitin ligase. It is essential for embryonic development in mammals[12,13]. UBR5 is frequently amplified and overexpressed in many cancer types, especially in human breast cancer and OC[14]. Our previous work has uncovered a distinctive role of UBR5 in the aggression of a murine triple negative breast cancer (TNBC) model[15]. Besides aberrant expression in TNBC, UBR5 was amplified in ~22% of OC in a TCGA (The Cancer Genome Atlas)-based analysis, and its expression in patient specimens was increased in 47% of OC tumor tissues examined[14,16]. UBR5 was described as an adverse prognostic factor for serous epithelial OC and able to confer cisplatin resistance on OC cell lines[16–18]. High UBR5 expression levels are strongly associated with poor OC patient survival[19]. However, mechanistic connections between UBR5 and OC progression and metastasis remain largely unknown.

In the present study, we show that UBR5$^-$ mediated immunosuppressive TAM infiltration via CCL2/CSF-1 enhances OC growth and metastasis. UBR5 is also required for sustaining β-catenin signaling to promote cell adhesion/colonization and spheroid formation in a cell-intrinsic manner. Targeting *Ubr5* not only impairs OC development, but also augments the therapeutic benefit of chemotherapy and immunotherapies with immune checkpoint blockade or chimeric antigen receptor (CAR)-T cells. These findings reveal a profound role of UBR5 both through paracrine action driving OC-immune cell crosstalk and through cell-autonomous facilitation of OC spheroid formation. UBR5 emerges as a therapeutic target for the deadliest gynecological malignancy.

## Results

### Attenuated OC growth and peritoneal implantation of UBR5$^-$ deficient ID8 tumor.

*UBR5* was amplified in ~22% of OC patients in a TCGA (The Cancer Genome Atlas) data set. UBR5 expression levels were higher in human OC specimens than in normal ovaries. High expression of UBR5 was associated with poorer patient prognosis and shortened survival rates (Supplementary Fig. 1). These data demonstrate strong clinical links of aberrant UBR5 statuses with OC disease states.

To investigate the functional importance of UBR5 in OC development and metastasis, we used a syngeneic murine OC model with ID8 cells expressing Muc16$^{ecto}$ (herein simply referred to as ID8), a glycoprotein upregulated in the majority of ovarian carcinomas and used as a serum biomarker for OC[20]. We first knocked out the endogenous *Ubr5* gene in ID8 cells via CRISPR/Cas9 and selected 3 "knockout" monoclones named ID8/*Ubr5*$^{-/-}$ (Fig. 1a and Supplementary Fig. 2a–c). As a control, ID8 cells were transfected with Cas9-GFP plasmid expressing a non-targeting guide sequence (the transfected cells are herein referred to as ID8/GFP). The ID8/*Ubr5*$^{-/-}$ monoclones displayed similar in vitro propagation capacities to that of ID8/GFP cells (Supplementary Fig. 2d). We then proceeded with one ID8/*Ubr5*$^{-/-}$ monoclone (#1) and compared its phenotypic features to those of ID8/GFP. The morphology of ID8/*Ubr5*$^{-/-}$ was altered from a cuboidal epithelial shape to an elongated mesenchymal shape and it became less "clustered" (Fig.1b). The altered morphology resembled "epithelial to mesenchymal transition" (EMT). We thus evaluated the migratory ability of ID8 *Ubr5*$^{-/-}$ cells in vitro in a double chamber assay and a wound healing assay, which showed that these cells acquired a greater mobility compared to ID8/GFP cells (Supplementary Fig. 2e, f). To assess the metastatic capacity of *Ubr5*$^{-/-}$ ovarian tumors, we intravenously injected ID8 into recipient mice. At 30 days post injection, we observed ~two times more ID8/*Ubr5*$^{-/-}$ tumor cells in the lungs compared with control tumor cells (Fig. 1c), but there were far fewer pulmonary and liver metastatic nodules in mice bearing ID8/*Ubr5*$^{-/-}$ at 60 days post injection (Fig. 1d and Supplementary Fig. 2g, h). Thus, ID8/*Ubr5*$^{-/-}$ failed to progress from lung micrometastases to macroscopic metastases. The expression of the epithelial marker β-catenin and cytokeratin 18 was decreased in ID8/*Ubr5*$^{-/-}$, whereas the EMT regulators zinc finger E-box binding homeobox 1 and 2 (ZEB1, 2) and mesenchymal marker Vimentin were upregulated (Fig. 1e and Supplementary Fig. 2i, j), indicating that *Ubr5* depletion compromised epithelial properties. Consistently, ID8/*Ubr5*$^{-/-}$ cells exhibited decreased ability to adhere to Matrigel and form clones (Supplementary Fig. 2k, l). These data suggest that loss of UBR5 impairs MET and the colonization properties of ID8 tumors in the lung.

We next implanted ID8 orthotopically into the ovarian bursa of syngeneic recipient mice. ID8/GFP bearing mice developed palpable solid tumor lesions 8 weeks after tumor inoculation, with gross hemorrhagic ascites, metastases at multiple peritoneal locations, as well as tumor-induced splenomegaly (Fig. 1f, g, Supplementary Fig. 3a–e and Supplementary Table 1). However, this process was markedly reduced in ID8/*Ubr5*$^{-/-}$ bearing mice and resulted in prolonged survival (Fig. 1h).Tumor progression was also inhibited in mice bearing ID8/*Ubr5*$^{-/-}$ upon intraperitoneal (i.p.) injection. These mice displayed dramatically attenuated tumor growth, impaired ascites accumulation, and diminished peritoneal implantation, compared with ID8/GFP bearing mice (Fig. 1i–k and Supplementary Fig. 3f). Consequently, survival was significantly enhanced in ID8/*Ubr5*$^{-/-}$ bearing mice with a median survival of 97 days compared to 51 days in control (Fig. 1l). It was worth noting that targeting *Ubr5* in ID8 didn't induce any pronounced pathogenic effects in mice (Supplementary Tables 1, 2).

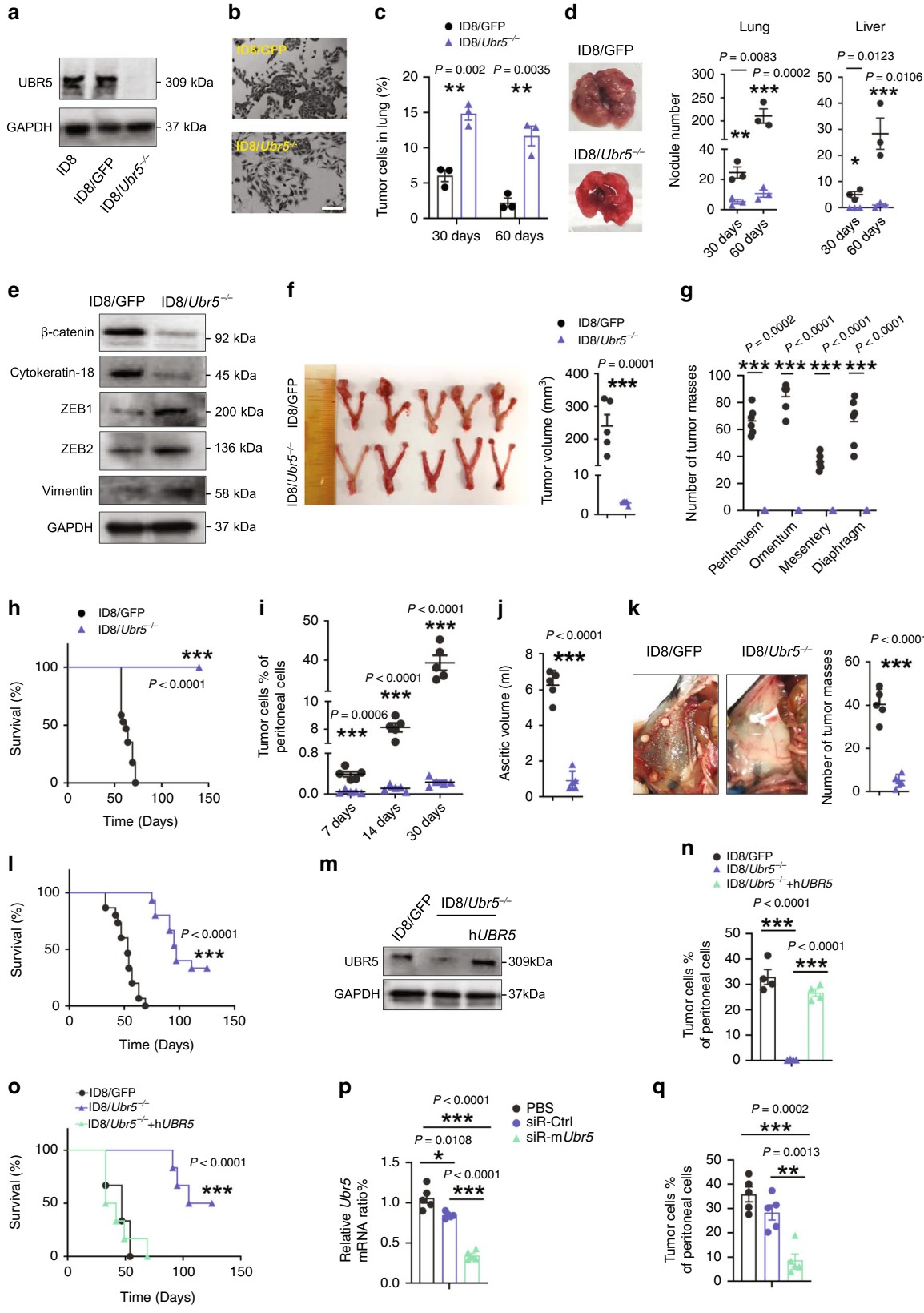

Importantly, reintroduction of ID8/$Ubr5^{-/-}$ tumors with the human orthologue of UBR5 rescued their defective OC growth (Fig. 1m–o). An alternative approach to CRISPR-mediated genetic deletion was taken to corroborate UBR5's tumorigenic activity by targeting UBR5 expression systemically using siRNA-containing nanoparticles in mice bearing ID8/GFP tumors. The effect of $Ubr5$ gene expression silencing was confirmed in tumor cells treated with nanoparticles (Fig. 1p). Administration of $Ubr5$-silencing nanoparticles strongly reduced the number of ID8 cells in the peritoneal cavity (Fig. 1q). Collectively, these data demonstrate a strong role of UBR5 in ID8 tumor growth.

**Fig. 1 Attenuated OC growth and peritoneal implantation of ID8/Ubr5$^{-/-}$ in mice. a** Deletion of *Ubr5* in ID8 was verified by Western blot.
**b** Representative micrographs of ID8 cell morphology. Scale bars: 200 μm. **c, d** To evaluate spontaneous metastasis, $5 \times 10^6$ ID8 cells were i.v. injected to C57BL/6 female recipient mice (*n* = 3 mice per group). **c** Quantification of Muc16$^+$ tumor cells. **d** Representative images (left) and quantified values (right) of metastatic nodules in lung or liver of recipient mice. **e** Protein expression of β-catenin, Cytokeratin-18, ZEB1, ZEB2, and Vimentin in ID8 cells was evaluated by western blot. **f** Tumors from orthotopic syngeneic model were resected and measured 8 weeks after tumor inoculation (*n* = 5 mice per group). **g** Metastatic nodules in peritoneum, omentum, mesentery and diaphragm were quantified (*n* = 6 mice per group). **h** Kaplan–Meier curves showing overall survival of mice (ID8/GFP *n* = 18, ID8/Ubr5$^{-/-}$ *n* = 13 mice per group), *P* < 0.0001, log-rank test. **i** Proportion of tumor cells in ascites at indicated times after tumor implantation (*n* = 5 mice per group). **j** Ascitic fluid volumes were measured at day 50 (*n* = 5 mice per group). **k** Peritoneal metastases were evaluated at day 50 (*n* = 5 mice per group). **l** Kaplan–Meier curves showing the survival rates of mice (*n* = 15 mice per group), *P* < 0.0001, log-rank test. **m** Protein expression of UBR5 in ID8/GFP, ID8/Ubr5$^{-/-}$, hUBR5 reintroduced ID8/Ubr5$^{-/-}$ cells was assessed by western blot. **n** Tumor burdens were evaluated at day 30 (*n* = 4 mice per group). **o** Survival rates were quantified (*n* = 6 mice per group), *P* < 0.0001, log-rank test. **p** mRNA expression of *Ubr5* in tumor cells was assessed. Data were normalized to puromycin N-acetyl- transferase (PAC) in each sample (*n* = 5 mice per group). **q** Proportions of Muc16$^+$CD45$^-$ tumor cells in ascites were quantified at day 30(*n* = 5 mice per group). All data are representative of at least two independent experiments with similar results. Data are shown as mean ± SEM, unpaired two-sided Student's *t*-test with no correction for multiple comparison, **P < 0.01, ***P < 0.001. Source data are provided as a Source Data file.

**Alterations in tumor microenvironment of ID8/Ubr5$^{-/-}$ tumor-bearing mice.** Immune cells that infiltrate the tumor microenvironment (TME) are co-opted by ovarian cancer cells to enable metastatic seeding and tumor growth. To examine the effect of tumor-derived UBR5 on various immune cells in TME, we analyzed the cellular composition in the peritoneal cavity. Although we observed a transient increase in infiltrating CD4$^+$ and CD8$^+$ T cells one-week post ID8/Ubr5$^{-/-}$ tumor implantation, there was little difference in T cell infiltration on day 14 and day 30 (Fig. 2a, b and Supplementary Fig. 4a, b). The proportion of Foxp3$^+$T regulatory cells changed little at various time points (Fig. 2c and Supplementary Fig. 4c). T cells are known to exert significant pressure against OC, and the presence of intra-tumoral T cells strongly correlates with improved outcome in advanced OC patients[21,22]. Hence, to determine whether the early T cell response induced by *Ubr5* deletion was responsible for the observed tumor diminution, we compared tumor growth in *Rag2$^{-/-}$* mice, CD4-deficient mice and CD8-deficient mice. Surprisingly, B and T cell deficiency did not mitigate the differences in growth capacity/survival between control and *Ubr5$^{-/-}$* tumors (Fig. 2d and Supplementary Fig. 4d). These results imply that the adaptive immune system is not actively involved in arresting *Ubr5$^{-/-}$* tumor progression. To determine whether NK cells participate in controlling *Ubr5$^{-/-}$* tumor, we performed antibody (anti-NK1.1 mAb) depletion of NK cells in ID8 bearing mice. Depleting NK cells did not restore ID8/Ubr5$^{-/-}$ tumor growth, suggesting that NK cells are not responsible for restraining *Ubr5$^{-/-}$* tumor (Supplementary Fig. 4e–h).

Myeloid cell populations within TME, including, tumor-associated dendritic cells (tDCs), monocytes, eosinophils, neutrophils, and tumor-associated macrophages (TAMs), were all decreased in ID8/Ubr5$^{-/-}$ bearing hosts (Supplementary Fig. 4i). Among them, infiltrating TAMs (CD11b$^+$, F4/80$^+$) displayed the most dramatic reduction, decreasing to under 15% of those in the peritoneal cavity of control mice (Fig. 2e). Reduced numbers of macrophages in the lung was also observed in mice injected *i.v.* with ID8/Ubr5$^{-/-}$ cells (Fig. 2f). Immunostaining confirmed that both CD68$^+$ TAMs and Ki67$^+$ cells from peritoneal ascites were greatly decreased in ID8/Ubr5$^{-/-}$ bearing mice (Fig. 2g–i). In addition, the number and size of spheroids also diminished with *Ubr5* deficiency (Fig. 2j). Together, these data suggest that tumor derived UBR5 plays a critical role in regulating macrophage recruitment into ID8 tumors.

**Essential role of TAMs in UBR5-regulated OC growth and spheroid formation.** To determine whether impaired ID8/Ubr5$^{-/-}$ tumor growth is a consequence of reduced macrophage recruitment, we isolated CD11b$^+$ F4/80$^+$ TAMs from ID8/GFP tumor bearing

donor mice, mixed them with ID8/Ubr5$^{-/-}$ tumor cells at 1:1 ratio and transferred the cells into naïve recipient mice. Mice injected with TAMs or ID8 cells alone served as controls (Supplementary Fig. 5a). Co-administration with TAMs remarkably boosted ID8/Ubr5$^{-/-}$ tumor growth, ascites fluid accumulation, tumor proliferation, and spheroid formation to the levels seen with the ID8/GFP tumor (Fig. 3a–c and Supplementary Fig. 5b–d). Accordingly, the survival of ID8/Ubr5$^{-/-}$ tumor bearing mice was greatly shortened by TAM complementation (Fig. 3d).

Conversely, depletion of macrophages in ID8 bearing mice with liposome-encapsulated clodronate (LC) markedly reduced tumor burden and increased mouse survival (Fig. 3e–h and Supplementary Fig. 5h–m). LC treatment depleted macrophages from the peritoneal cavity without affecting T cell accumulation. Nor did LC exert a direct anti-proliferative effect on tumor growth in vitro (Supplementary Fig. 5n–o). Thus, UBR5 deficiency strongly impairs TAM recruitment, an essential event that promotes OC growth. Nonetheless, LC depleted peritoneal TAMs in the ID8/GFP group to a level even lower than that in ID8/Ubr5$^{-/-}$ group (Fig. 3e and Supplementary Fig. 5i–k), but ID8/GFP bearing mice lacking TAMs still demonstrated a higher tumor burden (Fig. 3f) with more proliferative cells (Fig. 3g and Supplementary Fig. 5j), spheroid accumulation (Fig. 3h), and shorter survival (Fig. 3i) than the ID8/Ubr5$^{-/-}$ group without LC treatment. Thus, additional factors besides TAMs are involved in UBR5-regulated tumor-promoting activities.

**Altered immunosuppressive properties of TAMs in UBR5-deficient tumor.** To establish the cause of the defective macrophage recruitment in *Ubr5*-null tumors, we performed transwell assay with isolated peritoneal macrophages and cell-free supernatant derived from ID8/GFP or ID8/Ubr5$^{-/-}$ tumor cells. We observed a ~50% reduction of migrated macrophages in response to ID8/Ubr5$^{-/-}$ tumor supernatant (Fig. 4a), suggesting that UBR5 deficiency attenuates macrophage recruitment in a paracrine manner. TAMs are polarized to M2-like subtype in peritoneal microenvironment during OC progression, and contribute to OC proliferation and migration[23]. To determine whether OC-derived UBR5 could modulate TAM functionality, CD11b$^+$ F4/80$^+$ macrophages were retrieved and assessed. RNA-seq analysis of the TAMs from WT and *Ubr5$^{-/-}$* tumor bearing mice revealed two groups of genes of interests (Supplementary Fig. 6a). The first group of genes (G1) changed little in TAMs from WT tumor-bearing mice compared to naïve macrophages but were strongly induced in TAMs from *Ubr5$^{-/-}$* tumor-bearing mice. The second group of genes (G2) exhibited the reverse pattern (Supplementary Data 1, 2). An ingenuity pathway analysis (IPA) of this dataset further revealed some interesting molecular properties of these TAMs. Moreover,

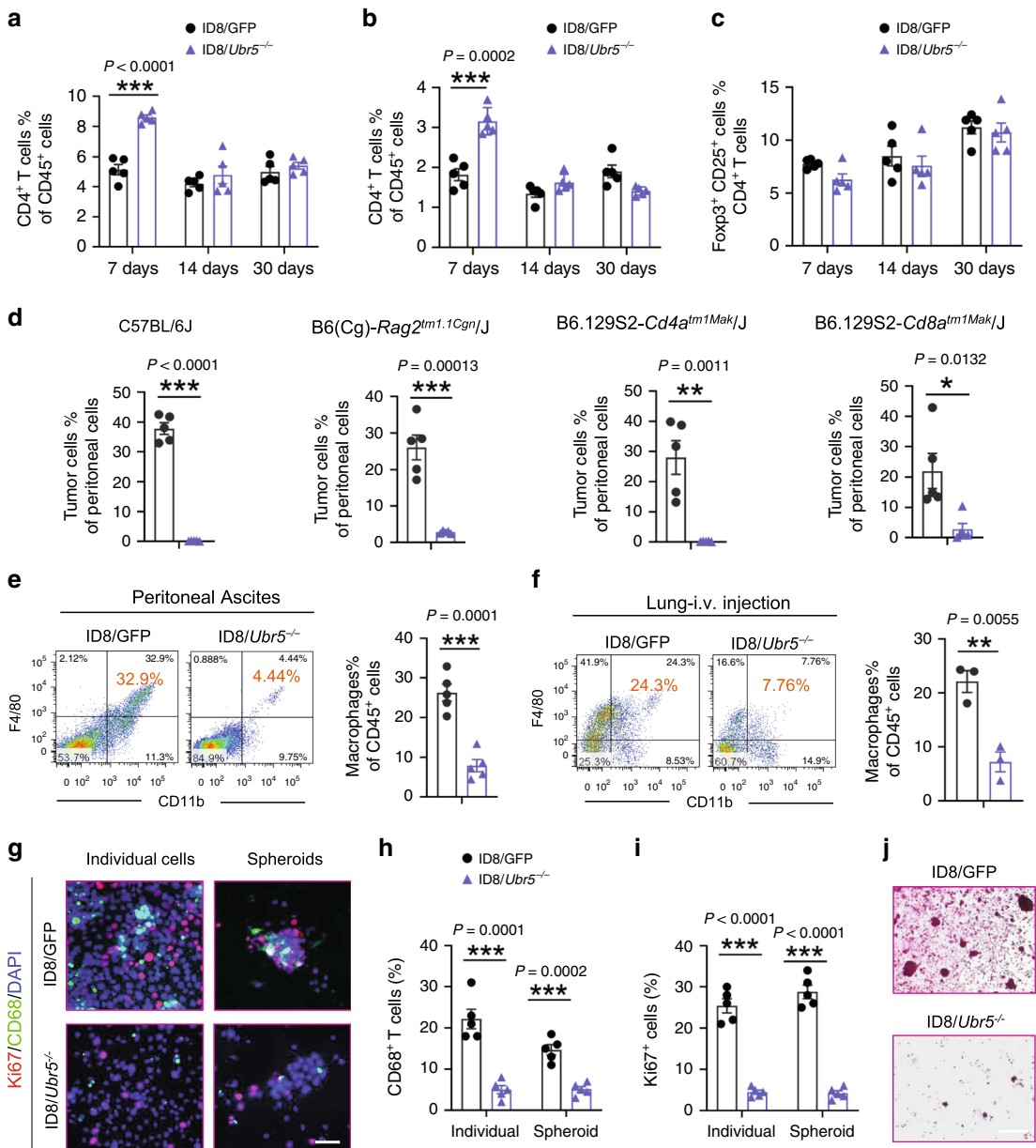

**Fig. 2 Analysis of immune cell involvement in regulating ID8/$Ubr5^{-/-}$ tumor growth. a–c** Peritoneal washes were collected at indicated times from OC bearing mice after peritoneal implantation of ID8 cells ($n = 5$ mice per group). CD4$^+$ (**a**) /CD8$^+$ (**b**) T cells were analyzed on CD45$^+$ infiltrating cells. **c** Foxp3$^+$CD25$^+$ regulatory T cells were analyzed on CD45$^+$CD3$^+$CD4$^+$ tumor infiltrating T cells (TILs). **d** B and T cell deficiency did not mitigate the differences in growth capacity between control and $Ubr5^{-/-}$ tumors. Proportion of Muc16$^+$ CD45$^-$ tumor cells in ascites were quantified at day 30 post OC implantation ($n = 5$ mice per group). **e, f** Representative FACS images of infiltrated CD11b$^+$F4/80$^+$ macrophages in peritoneal cavity or lung at day 30 after tumor injection. Proportion of macrophages on CD45$^+$ infiltrating cells in peritoneal cavity (**e**) ($n = 5$ mice per group), and lung (**f**) ($n = 3$ mice per group) were quantified. **g** Representative images of CD68$^+$ macrophages and surrounded Ki67$^+$ cells. All panels are the same magnification, scale bars: 50 μm. **h–i** CD68$^+$ cells (**h**) and Ki67$^+$ cells (**i**) in individual and spheroid populations were quantified ($n = 5$ mice per group and ten confocal images acquired from each sample). **j** At day 30 post $i.p.$ injection, spheroids from peritoneal washes were harvested and evaluated by H&E staining ($n = 5$ mice per group and an average of 10 fields acquired from each sample). Scale bars: 200 μm. In all cases, data are representative of at least two independent experiments with similar results. Data are presented as mean ± SEM, unpaired two-sided Student's t-test with no correction for multiple comparison, *$P < 0.05$, **$P < 0.01$, ***$P < 0.001$. Source data are provided as a Source Data file.

compared to TAMs from WT tumor-bearing mice, those from $Ubr5^{-/-}$ tumor-bearing mice exhibited higher levels of M1 macrophage markers (e.g., $il12a$, $Ccr2$) and lower levels of M2 markers (e.g., $Cx3cr1$, $il10$) (Fig. 4b and Supplementary Fig. 6b). Strikingly, expression of $Cx3cr1$, which is increased during monocyte maturation and inversely correlated with CCR2 in the blood[24], was markedly induced in TAMs from WT tumor-bearing mice while totally inhibited in the absence of UBR5. TAMs from ID8/$Ubr5^{-/-}$

bearing mice displayed strongly reduced levels of intracellular arginase-1 (Fig. 4c) and surface PD-L1 (Fig. 4d) expression in comparison with TAMs from ID8/GFP bearing mice. Although TAMs from both hosts were comparable in their ability to stimulate ID8 tumor migration (Supplementary Fig. 6c–e), TAMs from ID8/$Ubr5^{-/-}$ tumor-bearing mice exhibited weaker T cell proliferation-inhibiting activities than those from ID8/GFP tumor bearing mice (Fig. 4e), suggesting that tumor-derived UBR5 not only affects

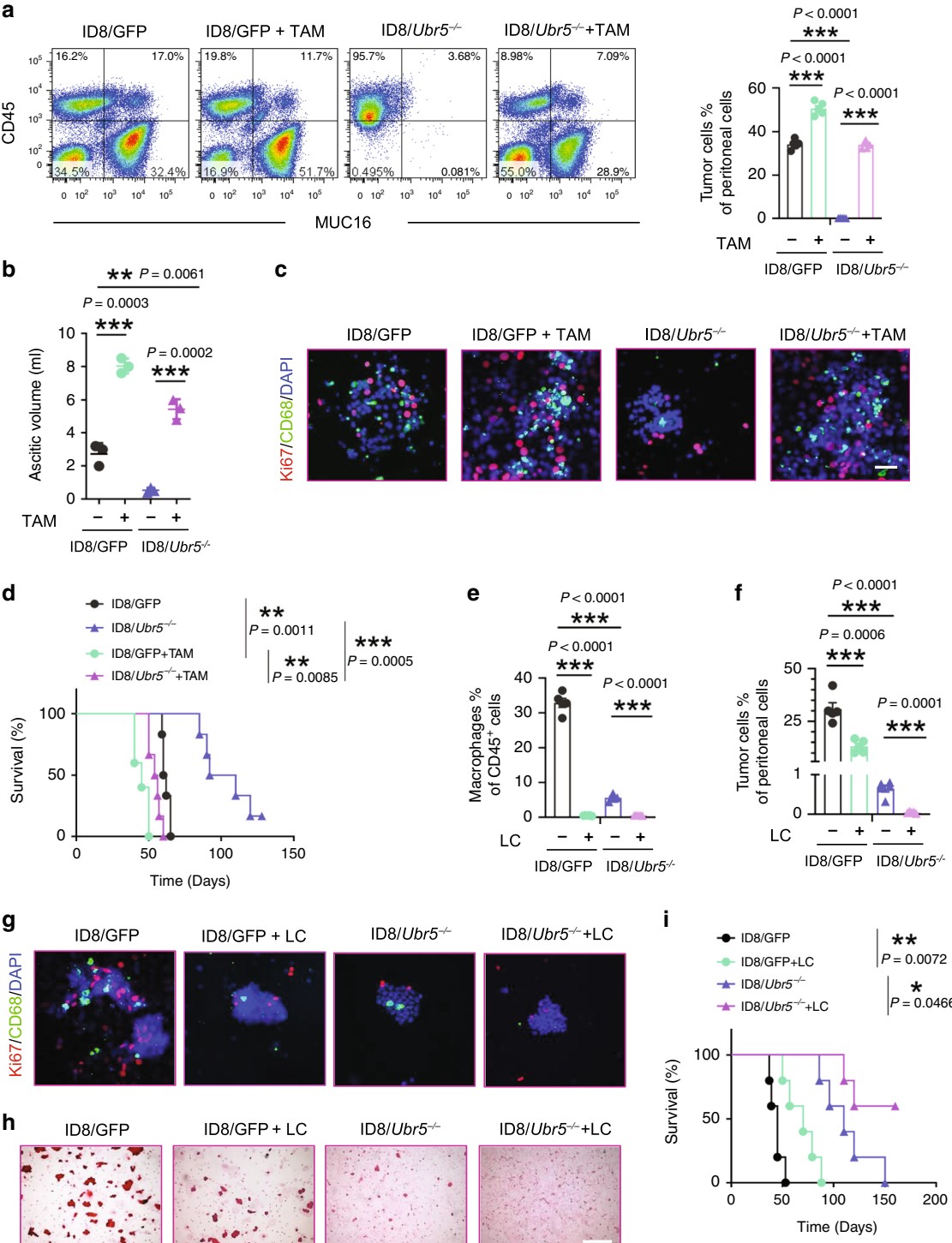

**Fig. 3 Impaired macrophage recruitment and attenuated peritoneal growth of UBR5-deficient OC. a** Representative FACS images and proportion of Muc16+CD45− tumor cells ($n = 5$ mice per group). **b** Ascitic fluid volumes were measured at day 40 ($n = 5$ mice per group). **c** Representative images of CD68+ macrophages and surrounded Ki67+ cells. All panels are the same magnification, scale bars: 50 μm. **d** Survival in ID8 bearing mice with or without exogenous TAMs ($n = 6$ mice per group), log-rank test. **e, f** Proportion of infiltrated CD11b+F4/80+ macrophages (**e**) and Muc16+ CD45− tumor cells (**f**) ($n = 5$ mice per group). **g** Representative images of CD68+ macrophages and surrounded Ki67+ cells ($n = 3$ mice per group and ten confocal images acquired from each sample). All panels are the same magnification, scale bars: 50 μm. **h** Spheroids from peritoneal washes (same volume) were harvested and evaluated by H&E staining ($n = 3$ mice per group and an average of ten fields acquired from each sample). Scale bars: 200 μm. **i** Survival in ID8 bearing mice with or without LC treatment ($n = 5$ mice per group), log-rank test. All data are representative of two independent experiments with similar results. Data are presented as mean ± SEM, unpaired two-sided Student's $t$-test with no correction for multiple comparison,*$P < 0.05$, **$P < 0.01$, ***$P < 0.001$. Source data are provided as a Source Data file.

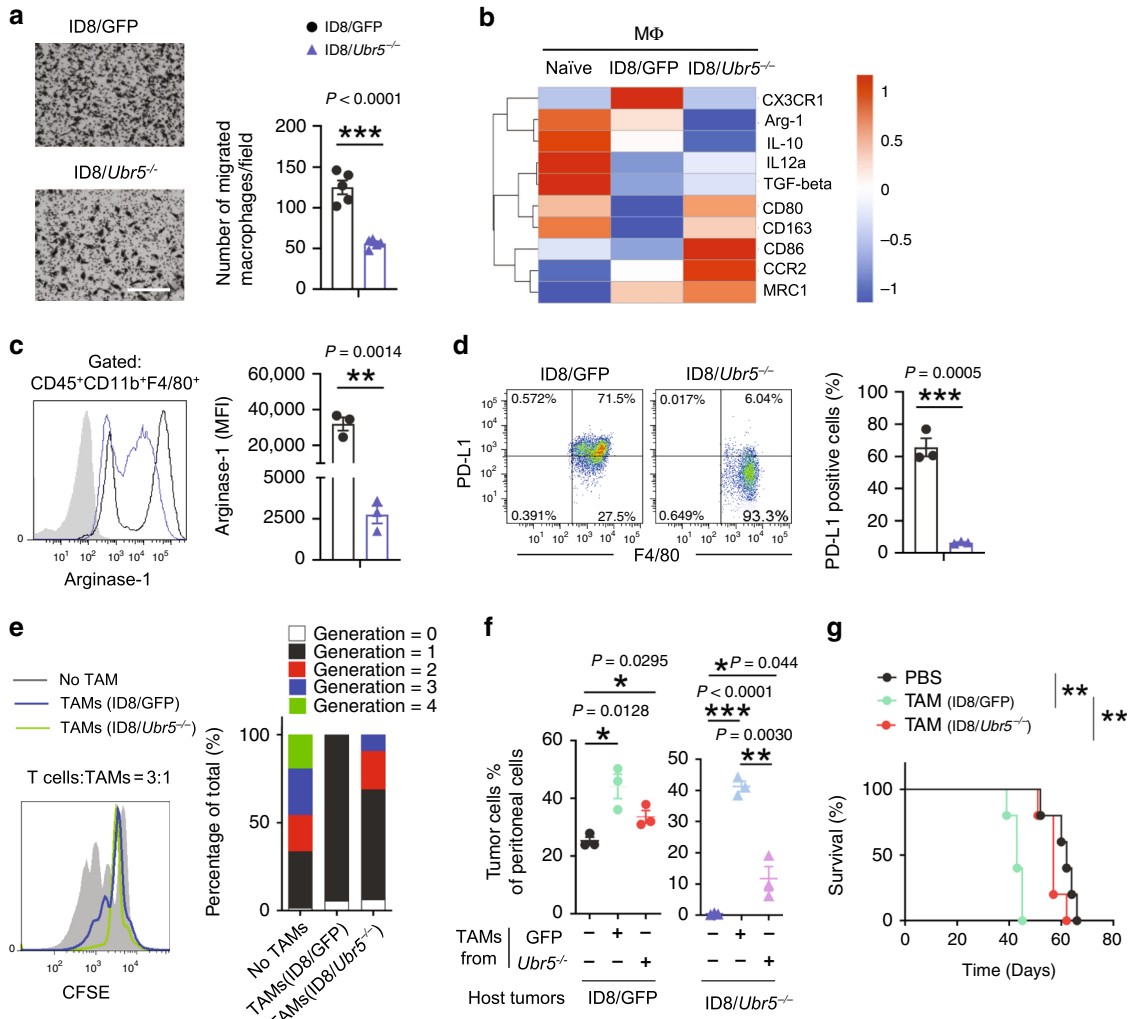

**Fig. 4 Functional impairment of macrophages in *Ubr5*-null tumors. a** Peritoneal macrophages were isolated and starved overnight before loaded into upper chamber with ID8 cells cultured at the bottom. Migrated cells were assessed after 16 h incubation ($n = 5$ biologically independent samples per group and an average of five fields acquired from each sample). Scale bars: 200 μm. **b** Heat map representation of differentially expressed M1/M2 genes in naïve macrophages and TAMs. **c** Representative FACS analysis of intracellular Arginase-1 in CD45$^+$ CD11b$^+$ F4/80$^+$ cells and quantification expressed as mean florescence intensity (MFI) of Arginase-1 ($n = 3$ mice per group). **d** Representative FACS analysis and proportion of PD-L1$^+$ cells gated on CD45$^+$ CD11b$^+$F4/80$^+$ cells ($n = 3$ mice per group). **e** T cell proliferation suppression assay. CFSE-labeled T cells from naïve mice were stimulated with anti-CD3 and CD28 antibodies and co-cultured for 3 days with TAMs isolated from ID8/GFP or ID8/*Ubr5*$^{-/-}$ bearing mice at 3:1 ratio ($n = 3$ mice per group). **f** Proportion of Muc16$^+$CD45$^-$ tumor cells with or without exogenous TAMs on day 45 post-tumor inoculation ($n = 3$ mice per group). **g** Kaplan–Meier curves showing the survival of ID8 bearing mice with or without exogenous TAMs ($n = 5$ mice per group), $P = 0.0026$, log-rank test. Data are representative of two independent experiments with similar results. Data are presented as mean ± SEM, unpaired two-sided Student's $t$-test with no correction for multiple comparison,*$P < 0.05$, **$P < 0.01$, ***$P < 0.001$. Source data are provided as a Source Data file.

TAM recruitment to TME in a paracrine manner but also TAMs' immunosuppressive activity. Indeed, when adoptively transferred to tumor-bearing hosts, TAMs from ID8/GFP tumor-bearing mice strongly promoted the growth of control and *Ubr5*$^{-/-}$ tumors, whereas TAMs from *Ubr5*$^{-/-}$ tumor-bearing mice did so poorly (Fig. 4f). Consequently, TAMs from ID8/GFP tumor-bearing mice shortened the survival of the recipient mice, whereas those from *Ubr5*$^{-/-}$ tumor-bearing mice did not (Fig. 4g). These data indicate that UBR5 deficiency in the tumor compromises TAMs' infiltration and impairs their immunosuppressive properties in a paracrine manner.

**Intrinsic defects in spheroid formation in *Ubr5*-deficient tumor**. We found that *Ubr5*-deficient ID8 cells expanded more slowly in both 3D spheroid cultures (Fig. 5a), and 3D coculture system with TAMs (Fig. 5b, c), revealed by the decreased number

and size of spheroids (Supplementary Fig. 7a, b). Of note, TAMs from ID8/*Ubr5*$^{-/-}$ bearing mice exhibited a similar capacity to those from control mice to facilitate spheroid formation, suggesting that UBR5 regulates ovarian tumor cell adhesion and spheroid formation in a cell-intrinsic manner independently of TAM-derived factors. In addition, apoptosis-suppressing effects of UBR5 have been documented in several cancer cell lines[18]. We observed that in vitro, *Ubr5* depletion in ID8 cells resulted in elevated PARP expression and correspondingly increased PARP cleavage, which is considered to be hall mark of apoptosis (Fig. 5d) but the ratio of cleaved vs. total PARP was not altered. In vivo, ID8/*Ubr5*$^{-/-}$ tumor bearing mice didn't exhibit enhanced TUNEL-positive signals in lung with intravenous injection, compared to control group (Fig. 5e). These data indicate that apoptosis is unlikely a major cause of impaired tumorigenesis of ID8/*Ubr5*$^{-/-}$. UBR5 has been reported to be a regulator of the

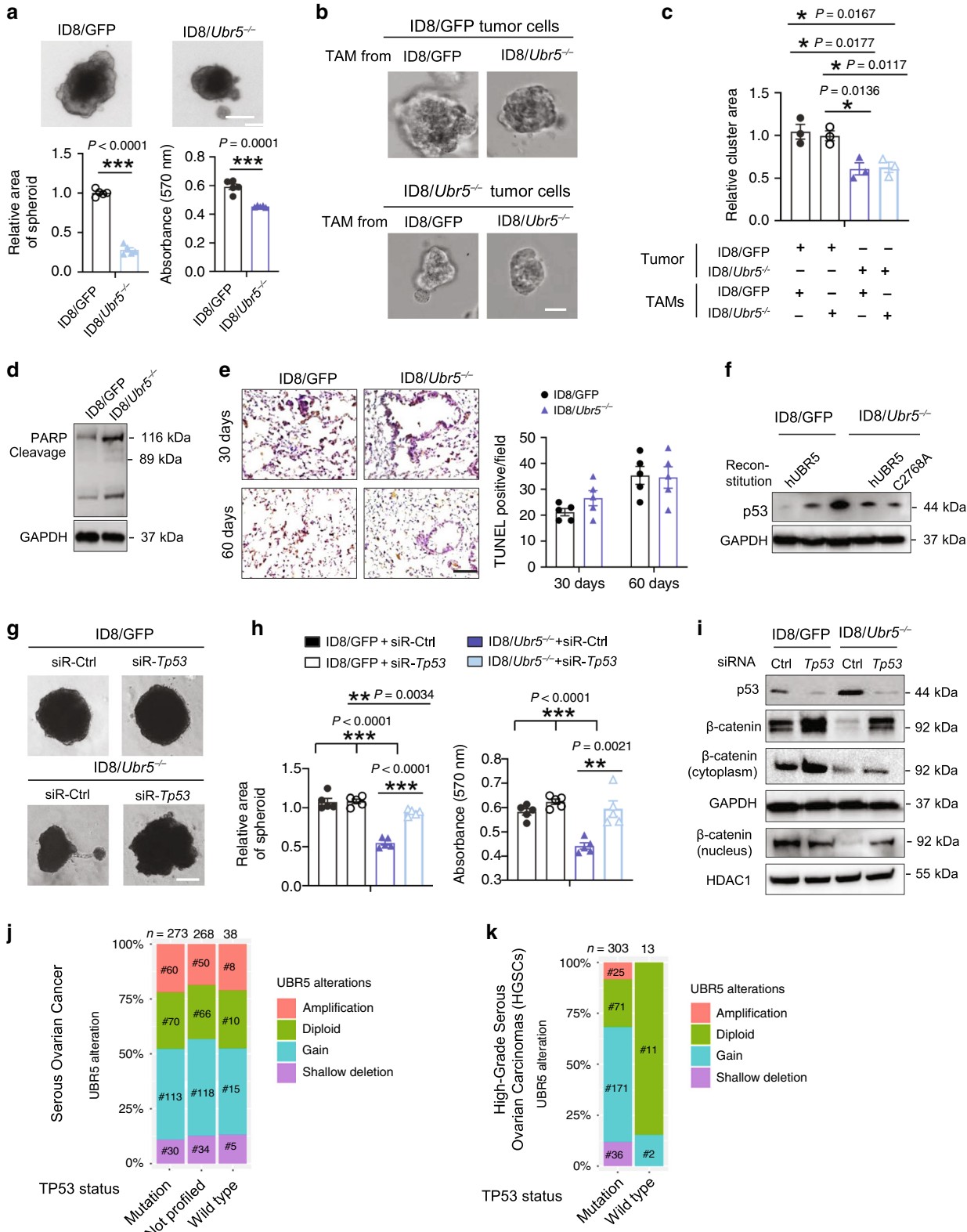

p53 protein, a tumor suppressor[25,26]. Compromised p53 expression and loss of function mutations in *TP53* contribute to the generation of cancer stem cells (CSCs)[27–29].

Consistently, we observed an elevated p53 protein level in ID8/*Ubr5*[-/-] tumor compared to WT tumor, whereas in UBR5-overexpressing ID8/GFP tumor, p53 level was decreased (Fig. 5f). Knocking down p53 in ID8/*Ubr5*[−/−] cells made them grow faster

in both 2D and 3D cultures (Fig. 5g, h and Supplementary Fig. 7c), suggesting that UBR5 promotes OC progression at least in part by regulating the level of p53. It is noteworthy that knocking down p53 expression in ID8/*Ubr5*[−/−] cells restored β-catenin to the control level associated with more nuclear accumulation, which is indicative of enhanced transcriptional activity of β-catenin (Fig. 5i, and Supplementary Fig. 7d).

**Fig. 5 Impaired spheroid formation mediated by p53 in *Ubr5*-deficient tumor. a** Spheroid cell proliferation assay with quantitative analysis ($n = 5$ biologically independent samples per group). Scale bars: 25 μm. **b**, **c** Representative images of single spheroid, Scale bars: 50 μm, the size of spheroids were statistically evaluated ($n = 3$ biologically independent samples per group and an average of five fields acquired from each sample). **d** Western blotting showing the expression of apoptosis-associated protein PARP, cleaved-PARP in ID8. **e** Apoptosis in lung sections was evaluated by TUNEL staining at indicated time post i.v. injection of ID8 ($n = 5$ mice per group and an average of five fields acquired from each sample), Scale bar: 400 μm. **f** Western blot showing p53 protein expression in indicated ID8 cells. **g**, **h** Spheroid cell proliferation assay with quantitative analysis ($n = 4$ biologically independent samples per group). Scale bars: 25 μm. **i** Western blot showing that knocking down *Tp53* in ID8 cells increased β-catenin expression. **j**, **k** The relationship between *TP53* mutation and *UBR5* gene alterations in serous ovarian carcinoma (**j**) and high-grade serous carcinoma (**k**) cohorts from the TCGA databases. Data are representative of two independent experiments with similar results. Data are presented as mean ± SEM, unpaired two-sided Student's *t*-test with no correction for multiple comparison, \*$P < 0.05$, \*\*$P < 0.01$, \*\*\*$P < 0.001$. Source data are provided as a Source Data file.

Approximately 96% of high-grade serous ovarian carcinomas (HGSCs), the most common and deadliest type of ovarian cancer, harbor *TP53* mutations. We further delineated the relationship between *Tp53* mutation status and *UBR5* alterations in a TCGA cohort, which includes about 600 serous ovarian cancer samples. Similar ratios of *UBR5* alterations among *TP53* wild type, mutation, and not profiled subgroups were observed (Fig. 5j). However, in a HGSC cohort[30], we identified a close association between *UBR5* alterations and *TP53* mutations, i.e., cancers with *UBR5* gene alterations, predominantly amplification and gains, harbored frequent *TP53* mutations, whereas those in the diploid state (without *UBR5* alterations) were mostly without *TP53* mutations (Fig. 5k), suggesting a possible regulatory relationship between *UBR5* alterations and *TP53* mutations. Together, these data demonstrate that UBR5 regulates β-catenin signaling via a p53-mediated pathway.

**Impaired cytokine/chemokine production caused by UBR5 deficiency.** To further elucidate the molecular mechanisms involved in UBR5-regulated paracrine activities, we performed RNA-seq analyses with retrieved peritoneal cells from ascites and found that genes involved in macrophage recruitment (such as *Ccl2, Vegf, Il6, Csf-1*, and *Cxcl1*) were significantly downregulated in peritoneal TME of ID8/*Ubr5*$^{-/-}$ bearing mice (Fig. 6a). Genes in *Wnt* signaling, growth factors were also downregulated by UBR5 deficiency. Many of them such as platelet-derived growth factor (PDGF), epidermal growth factor receptor (EGFR), vascular endothelial growth factor (VEGF) α, β, and fibroblast growth factor receptor (FGFR) are strongly involved in tumor stroma functions, vascular endothelial formation, and angiogenesis[31]. The broad downregulation of gene expression in the TAM infiltration/differentiation pathway (including *Ccl2, Csf-1, Cxcl1, and Il6*) was further confirmed in ID8/*Ubr5*$^{-/-}$ cells (Supplementary Fig. 8a). In particular, CCL2 and CSF-1 proteins were abundantly secreted by UBR5-sufficient tumors, while forced human UBR5 overexpression in ID8/GFP further increased their expression (Fig. 6b, c and Supplementary Fig. 8b). Cell adhesion molecules play important roles in OC spheroid formation[32]. Although UBR5 deficiency barely altered the levels of type I calcium-dependent cadherins (N-cadherin/E-cadherin) and EGFR/ICAM-1, which mediate OC spheroid compaction[23,33,34], it abrogated β-catenin expression in ID8 (Supplementary Fig. 8c). β-catenin is a cancer stem cell (CSC) marker enriched in OC spheroids, and activation of β-catenin regulates the tumor-initiating capacity and spheroid formation[35,36]. To evaluate the role of the UBR5-regulated paracrine factors CCL2, M-CSF, and β-catenin in macrophage activation and tumorigenesis, we expressed these three molecules in ID8/*Ubr5*$^{-/-}$ cells. Reintroducing CCL2, M-CSF, and β-catenin shifted ID8/*Ubr5*$^{-/-}$ cells to a more clustered epithelial shape and partially restored their abilities to generate spheroids, recruit macrophages and form colonies (Fig. 6d–f and Supplementary Fig. 8e–h). Accordingly, ID8/*Ubr5*$^{-/-}$ tumor growth in mice was partially complemented

by the reintroduction, with increased TAMs infiltration, cellular proliferation, spheroid accumulation, and shortened survival (Fig. 6g–k). Overexpression of UBR5 in ID8/GFP further augmented tumor progression (Fig. 6k), highlighting its oncogene-like activity in OC. As a transcriptional co-activator, β-catenin is reported to play a role in transactivation of CCL2[37]. Consistent with this role, reintroduction of β-catenin in both ID8/GFP and ID8/*Ubr5*$^{-/-}$ induced higher levels of CCL2 expression (Supplementary Fig. 8i). Further, *Tp53* knocking down in ID8 cells resulted in increased β-catenin expression and CCL2 production (Supplementary Fig. 8j), indicating an UBR5 dependent p53-β-catenin-CCL2 axis in ID8. Taken together, these results indicate that paracrine factors (CCL2/M-CSF) and cell-intrinsic β-catenin signaling contribute to UBR5-mediated OC progression.

**Requirement of UBR5 for human ovarian cancer growth and macrophage recruitment.** Several lines of investigation suggested that the above findings in mice are likely to be relevant in human OC. Immunohistochemical (IHC) staining for UBR5, CD68, and Ki67 in primary tumors of 50 OC patients showed a positive correlation between UBR5 expression and CD68$^+$ macrophage infiltration ($R = 0.6918$, $P < 0.0001$), as well as between UBR5 expression and the density of proliferating, Ki67$^+$ cells ($R = 0.2105$, $P = 0.001$) (Fig. 7a–c). Next, we investigated the functional role of UBR5 in human OC SKOV3 cells xenografted in Scid-Beige mice. Like murine OC, human SKOV3/*UBR5*$^{-/-}$ displayed decreased expression of the adhesion-related molecule β-catenin, the chemokines CCL2 and CXCL1, and the cytokines M-CSF and IL6) (Fig. 7d and Supplementary Fig. 9a, b). In vitro, compared to normal human ovarian surface epithelial cells HOSEpiC, SKOV3 cells proliferated more rapidly, migrated faster and form larger spheroid in 3D culture (Fig. 7d, e and Supplementary Fig. 9c, d). Targeting *UBR5* in SKOV3 further enhanced cell migratory property, but reduced spheroid proliferation ability and impaired clonogenic capacity, compared with control SKOV3 (Fig. 7e and Supplementary Fig. 9d, e). Similar alterations were also observed in high-grade serous ovarian cancer (HGSOC) cell line OVCAR3 with *UBR5* depletion (Supplementary Fig. 9f–j).

In vivo, SKOV3/*UBR5*$^{-/-}$ tumor bearing mice displayed reduced peritoneal tumor burden with impairment in peritoneal macrophage infiltration, spheroid formation, and cancer cell proliferation compared to controls (Fig. 7f–i). Accordingly, mouse survival was improved by deletion of *UBR5* in SKOV3 (Fig. 7j). However, the survival of SKOV3/*UBR5*$^{-/-}$ tumor bearing mice was markedly shortened to the control level when co-injected with TAMs isolated from SKOV3 tumor bearing donor mice (Fig. 7k). Collectively, these results indicate that tumor-derived UBR5 is required for human OC progression by molecular mechanisms similar to those in the mouse.

**Enhanced OC therapeutic outcomes via targeting tumor-derived UBR5.** Spheroids represent an invasive and chemoresistant

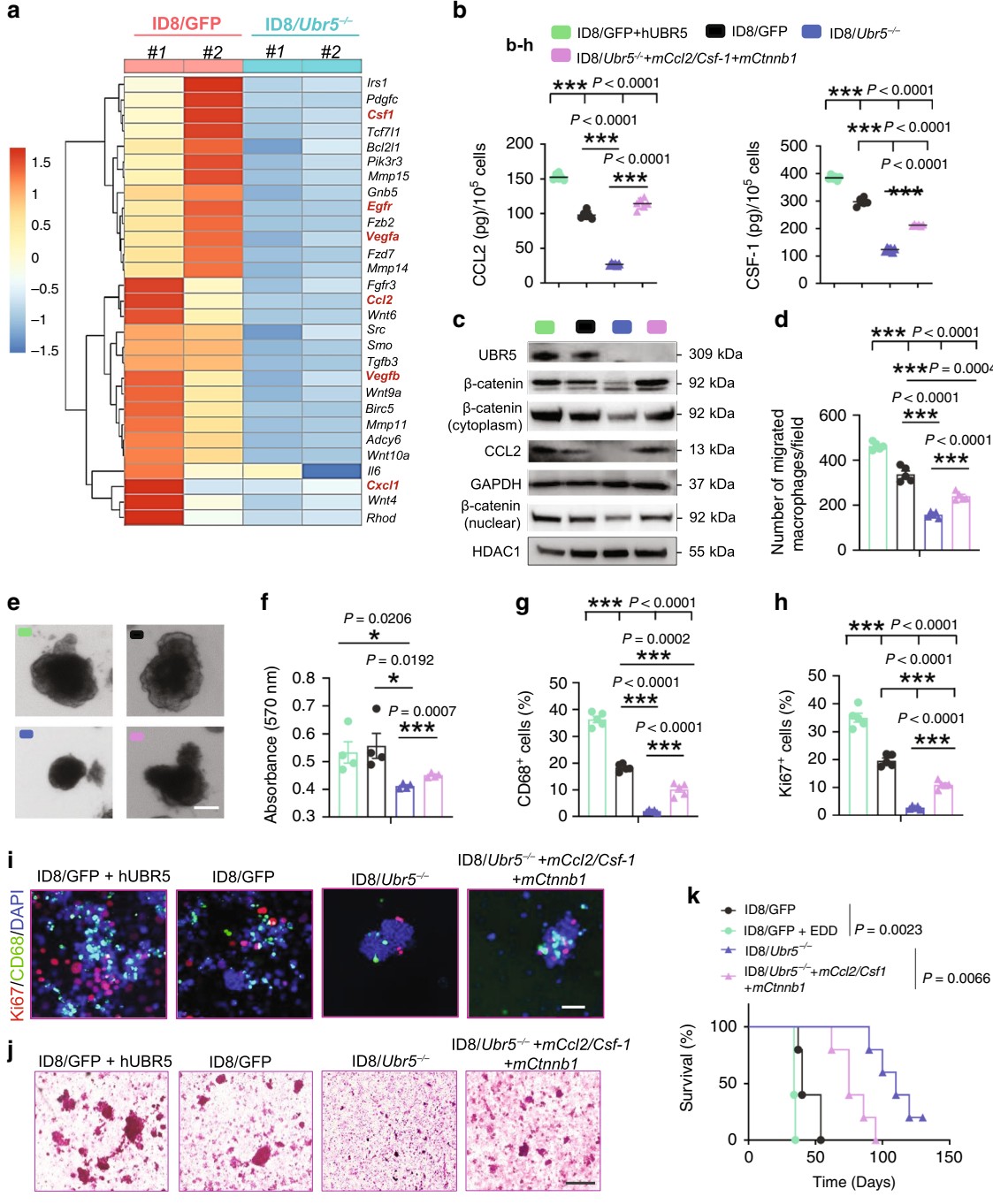

**Fig. 6 Effects of tumor-derived paracrine factors on TAMs and spheroids. a** Heat map representation of differentially expressed genes involved in macrophage recruitment, Wnt signaling, and cell adhesion in recovered peritoneal cells from ID8 bearing mice. **b** ID8 cells were cultured in DMEM overnight and the protein levels of CCL2 and CSF-1 in cell cultures were measured by ELISA ($n = 5$–6 biologically independent samples per group). **c** Western blotting showing the protein expression of UBR5/β-catenin/CCL2 in different ID8 cells. **d** Peritoneal macrophages were harvested and seeded into Transwell inserts with ID8 cell cultured in the bottom. Migrated macrophages were quantified after 16h-incubation ($n = 5$ biologically independent samples per group). **e** Representative images of spheroids. Scale bars: 200 μm. **f** Quantification of spheroid expansion rates ($n = 4$ biologically independent samples per group). **g**, **h** CD68$^+$ cells and Ki67$^+$ cells in spheroids were quantified ($n = 4$ mice per group). **i** Representative images of CD68$^+$ macrophages and surrounded Ki67$^+$ cells. ($n = 4$ mice per group and ten confocal images acquired from each sample). All panels are the same magnification, scale bars: 50 μm. **j** Spheroid retrieved from ID8 bearing mice at day 30 were underwent H&E staining ($n = 4$ mice per group and an average of five fields acquired from each sample). Scale bars: 200 μm. **k** Survival in mice bearing ID8 tumors ($n = 5$ mice per group), log-rank test. All data are representative of at least two independent experiments with similar results. Data are presented as mean ± SEM, unpaired two-sided Student's $t$-test with no correction for multiple comparison, $*P < 0.05$, $**P < 0.01$, $***P < 0.001$. Source data are provided as a Source Data file.

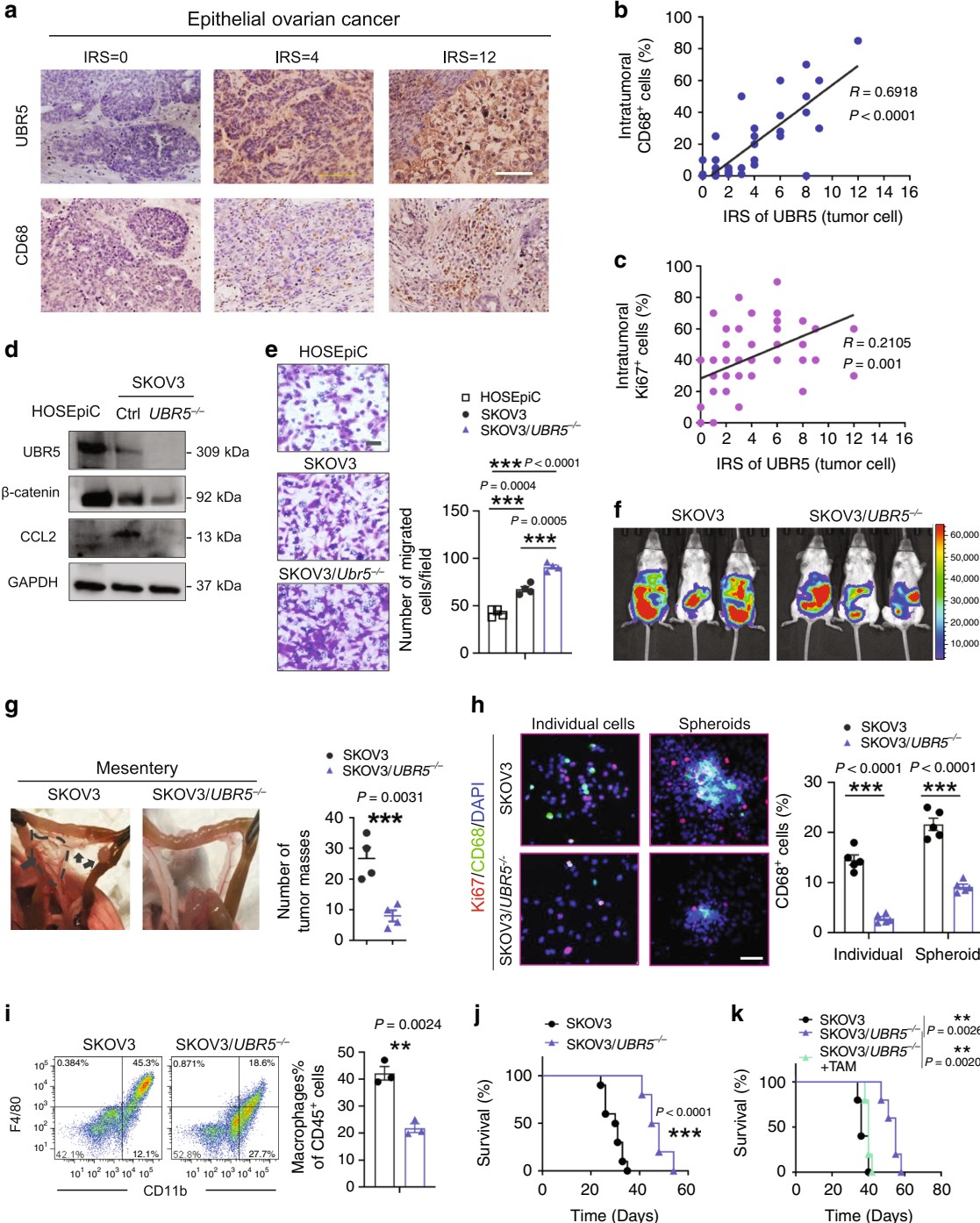

**Fig. 7 Effects of human OC-derived UBR5 on TAMs and tumor growth. a** IHC staining of UBR5 and CD68 in primary tumors from OC patients ($n =$ 50 samples). Scale bars: 200 μm. **b, c** Correlation assessed by Pearson correlation analysis and linear regression analysis between tumor UBR5 expression levels and intratumoral CD68[+] (**b**), Ki67[+] cell density (**c**) in human EOC patients ($n = 50$ samples). **d** Deletion of *UBR5* in SKOV3 cells was verified by western blot. Reduced β-catenin and CCL2 expression in SKOV3/*UBR5*[−/−] were determined at protein level, human ovarian surface epithelial cells HOSEpiC were used as control. **e** Representative micrographs and quantitation of Transwell-migration assay for SKOV3 ($n = 4$ biologically independent samples per group and an average of five fields acquired from each sample). Scale bars: 50 μm. **f** On day 21 after tumor implantation, imaging with Luciferase in SKOV3 tumor bearing SCID/Beige mice ($n = 3$ mice per group). **g** Representative images and statistical analysis of tumor implantation in mesentery ($n = 4$ mice per group). **h** Representative images of CD68[+] macrophages and surrounded Ki67[+] cells from peritoneal washes ($n = 4$ mice per group and ten confocal images acquired from each sample). All panels are the same magnification, scale bars: 50 μm. **i** Representative FACS images and quantification of infiltrated CD11b[+]F4/80[+] macrophages in peritoneal cavity ($n = 3$ mice per group). **j** Kaplan–Meier curves showing the survival of SKOV3 tumor-bearing mice ($n = 10$ mice per group). $P < 0.0001$, log-rank test. **k** Survival in mice bearing SKOV3/*UBR5*[−/−] with or without TAMs isolated from SKOV3 bearing donor mice were quantified ($n = 5$ mice per group), log-rank test. In all cases, data are representative of two independent experiments. Data are presented as mean ± SEM, unpaired two-sided Student's *t*-test with no correction for multiple comparison, **$P < 0.01$, ***$P < 0.001$. Source data are provided as a Source Data file.

cellular component in OC and the Wnt/β-catenin pathway has been shown to play critical role in chemoresistance[38,39]. Given that UBR5 promotes spheroid formation and enhances Wnt/β-catenin signaling, we reasoned that OC-derived UBR5 may contribute to chemoresistance. In vitro, overexpression of UBR5 increased resistance to cisplatin-induced cell death in OC (Fig. 8a), while ID8/*Ubr5*[-/-] were more susceptible to cisplatin-induced apoptosis (Fig. 8b). In vivo, UBR5-overexpressing ID8 grew more aggressively in mice and caused rapid death (Fig. 8c, d). The more rapid tumor growth of UBR5-overexpressing ID8 caused decreases in red blood cell count, hemoglobin, hematocrit and reticulocytes compared to the other two groups, indicating anemia (Supplementary Table 2). Moreover, UBR5 overexpression largely abrogated the therapeutic effect of cisplatin treatment. These results indicate that UBR5 contributes to chemoresistance in OC.

Immunotherapy has revolutionized the treatment for many types of cancer, but has had little clinical impact in OC. The modest responses in OC are attributed to the suppressive ascitic microenvironment, such as M2-polarized TAMs and Treg[9]. We have demonstrated that targeting tumor-derived UBR5 reverses the suppressive TME by impairing TAM recruitment, which can potentially reinforce the efficacy of immunotherapy. To test this hypothesis, we administered anti-PD-1 mAb to mice with established ID8 ovarian tumors. Although ID8-bearing mice were non-responsive to PD-1 blockade alone, *Ubr5* depletion synergized with PD-1 blockade to extend their survival (Fig. 8e).

To determine whether targeting UBR5 could enhance the benefit of adoptive T cell therapy, OC bearing mice were treated with 4H11m28mZ CAR-T cells targeting MUC16[ecto] expressed by ID8. 19m28mZ CAR-T cells targeting CD19 served as a specificity control[40]. Treatment of ID8/GFP bearing mice with 4H11m28mZ CAR T cells resulted in greater survival than seen with mice that received control CAR-T cells. Strikingly, *Ubr5*-targeted deletion combined with administration of 4H11m28mZ CAR T cells resulted in long-term survival (>250 days) of all the treated mice, highlighting synergy between UBR5 inhibition and specific CAR-T therapy (Fig. 8f). Together, these data demonstrate that targeting OC-derived UBR5 can alleviate chemoresistance and synergize with immunotherapies, strongly enhancing the therapeutic benefits of current OC treatments.

## Discussion

This study revealed that tumor derived UBR5 is a pivotal driver of OC aggressiveness. Depletion of *Ubr5/UBR5* in both mouse ID8-Muc16[ecto] OC and human SKOV3 OC blocked tumor growth and peritoneal metastasis by disrupting paracrine regulation of TAM infiltration and cell-intrinsic regulation of spheroid formation (Supplementary Fig. 10). Targeting *Ubr5* in ID8 OC synergistically improved therapeutic outcomes of chemotherapy and immunotherapies with immune-checkpoint blockade and CAR T cell administration. This work has elucidated the causality and mechanisms of UBR5's tumorigenic activities in OC and broadened our understanding of the biology of a novel E3 ligase in regulating cancer-immune cell crosstalk. Reducing the expression of UBR5 could be an efficacious therapeutic approach for OC.

Unlike many other epithelial cancers, malignant OC exfoliates as single cells and aggregates as multicellular spheroids in the peritoneal cavity, resisting anoikis. TAMs have been demonstrated to promote OC spheroid formation. In the present study, ID8/*Ubr5*[-/-] bearing mice displayed dramatically attenuated tumor growth with diminished peritoneal infiltration of myeloid cell populations, especially CD11b[+] F4/80[+] macrophages. Macrophage mobilization into OC TME is regulated by multiple

cytokine/chemokine and growth factors. Gene profiling identified downregulation of transcripts for chemokines and cytokines involved in macrophage recruitment (such as CCL2, CXCL1, and CSF-1) to ID8/*Ubr5*[-/-] TME. CCL2 and CSF-1 are important for inflammatory monocyte recruitment into the tumor site and differentiation into TAMs. Directly or indirectly, UBR5 transcriptionally regulates CCL2/CSF-1. It will be of great interest to understand how this regulation is affected. Given that UBR5 didn't modulate their mRNA stability, UBR5 may stabilize transcription factors (TFs) that control *Ccl2/Csf1* gene transactivation or act as coactivator of these TFs[41].

Interestingly, our findings imply an UBR5 orchestrated p53-β-catenin-CCL2 regulatory axis (Supplementary Fig. 8i, j), which warrants further exploration. We found that impaired tumorigenesis due to UBR5 deficiency could be fully restored by adoptive transfer of TAMs, indicating that the macrophage recruitment defect caused by *Ubr5* depletion mostly accounts for the decreased tumor burden and prolonged survival. This conclusion was further reinforced by our observation in human OC samples identifying a positive correlation between UBR5 expression and CD68[+] macrophage infiltration. In addition, when injected in mice, human OC SKOV3 xenografts with silenced UBR5 expression displayed similarly reduced peritoneal metastasis with less TAM infiltration than when UBR5 was expressed. Our study demonstrates the feasibility of targeting UBR5 as a novel approach to block TAM infiltration, an emerging intervention strategy against OC[42].

ID8/GFP bearing mice lacking TAMs still exhibited a higher tumor burden than ID8/*Ubr5*[-/-] bearing mice, implying that there are TAM-independent, cell intrinsic factors by which UBR5 contributes to OC progression. Indeed, loss of *Ubr5* impairs the adhesion ability and spheroid expansion capacity of ID8 and causes aberrant MET. β-catenin signaling is essential for maintaining stemness, driving aggressiveness and developing chemoresistance in OC[38,43]. Consistent with a report that UBR5 ubiquitinates β-catenin and subsequently enhances its stability[44], we found that depletion of *Ubr5/UBR5* in ID8 and SKOV3 decreased β-catenin levels, while overexpressing UBR5 in ID8 further enhanced β-catenin accumulation, demonstrating an oncogene-like property of UBR5.

Reintroduciton of *mCcl2, Csf-1*, and *mCtnnb1* into ID8/*Ubr5*[-/-] cells enhanced TAM recruitment, accelerated spheroid proliferation and colonization, and partially but significantly restored tumor growth. These data suggest that paracrine factors such as CCL2 and CSF-1, along with β-catenin signaling, contribute to the oncogenic functions of UBR5, although yet-to-be discovered, UBR5-dependent factors/pathways are also involved. UBR5 has been shown to suppress death receptor-induced apoptosis and down-regulate proapoptotic MOAP-1 in human OC cells[18,45]. However, we observed that PARP cleavage (Fig. 5d) was only marginally altered by *Ubr5* deficiency. A recent report showed that in human embryonic stem cells, loss of UBR5 triggered an increase in p53 levels and a concomitant decrease in cellular proliferation rates[46]. This finding is consistent with our own observation that UBR5 is a critical regulator of p53 protein, which mediates UBR5's organoid-forming activity and β-catenin expression, suggesting that UBR5-regulated p53 is an inhibitor of OC stemness.

Dysregulated E3 ligases employ diverse mechanisms to promote carcinogenesis by regulating a broad spectrum of substrates involved in DNA damage repair, apoptosis, tumor cell proliferation, and genomic instability[47]. However, most cancer-associated E3 ligases affect cell-intrinsic processes, and their roles in mediating the crosstalk between cancer and TME components have rarely been reported. We find that tumor-derived UBR5 promotes OC peritoneal implantation independent of the adaptive immune system and does so in a paracrine manner.

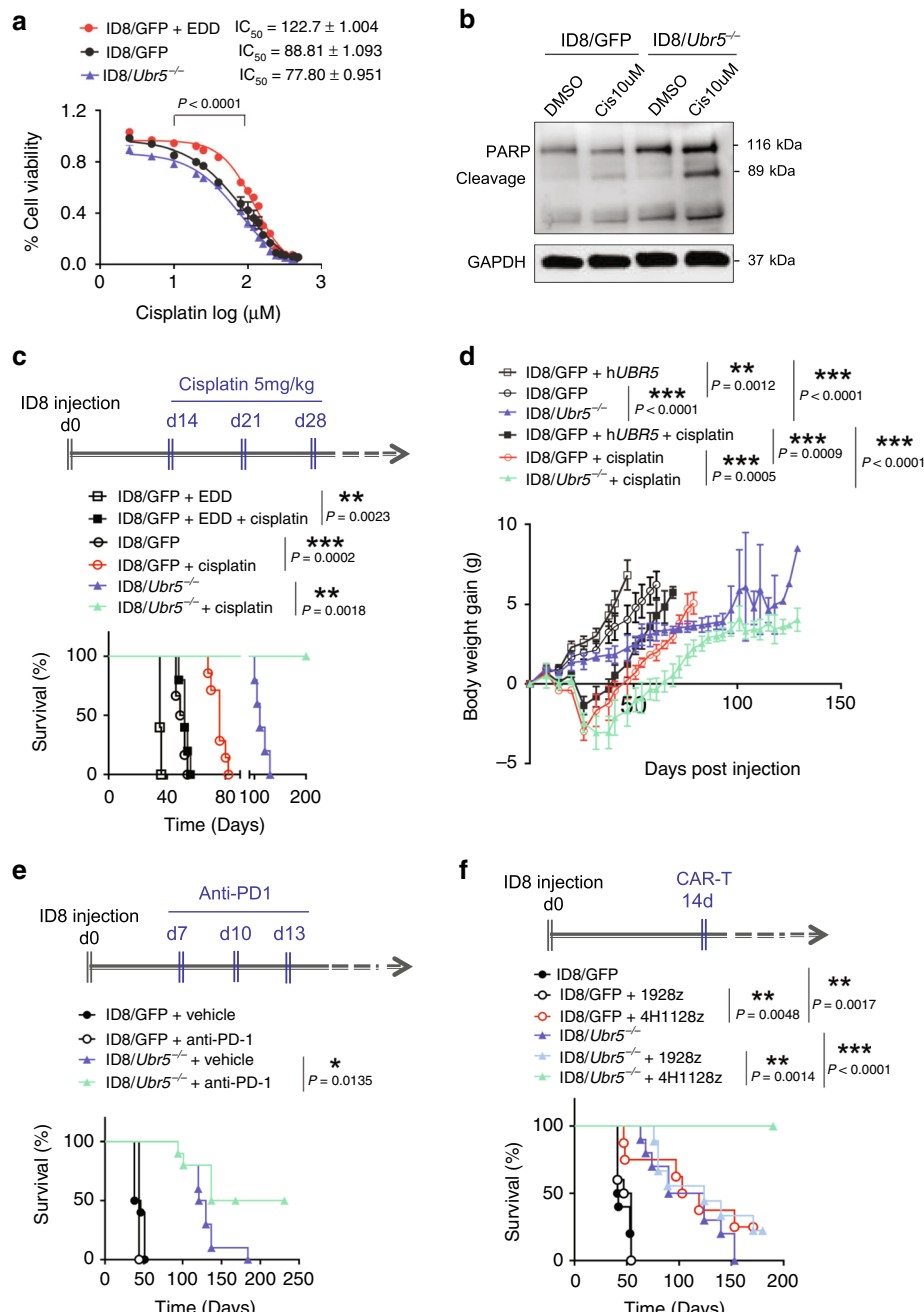

**Fig. 8 Enhanced therapeutic benefits in OC by targeting tumor-derived *Ubr5*. a** Dose response curves of ID8 treated with cisplatin for 72 h ($n = 5$ biologically independent samples per group). The averages of three independent experiments were plotted, data are presented as mean ± SD, $P < 0.0001$, one-way ANOVA test. **b** Western blot analysis of cleaved-PARP levels after 24 h of cisplatin (Cis) treatment in ID8 cells. **c**, **d** UBR5 overexpression abrogated the therapeutic effect of cisplatin treatment. ID8 bearing mice were treated with cisplatin at 2 weeks after tumor implantation once a week for 3 weeks ($n = 5$ mice per group). Mouse mortality (**c**) and body weight (**d**) were monitored. Data are presented as mean ± SD, unpaired two-sided Student's *t*-test. **e** Anti-PD-1 treatment enhanced the survival of ID8/*Ubr5*−/− bearing mice, but not ID8/GFP bearing mice. Mice were treated with anti-PD-1 at indicated times post tumor implantation, and survival rates were quantified ($n = 10$ mice per group). Data shown is pooled from two independent experiments. **f** 4H1128z cells administration resulted in long-term survival of mice harboring ID8/*Ubr5*−/−. Mice were treated with CAR-T cells 2 weeks after tumor implantation, and survival rates were quantified ($n = 10$ mice per group). In all cases, data are representative of at least two independent experiments. *P* values shown in **c**, **e**, and **f** were calculated by log rank test. *$P < 0.05$, **$P < 0.01$, ***$P < 0.001$. Source data are provided as a Source Data file.

Moreover, this is critical for tumor initiation and progression. Meanwhile, cell intrinsic regulation of cell adhesion, cell aggregation, and epithelial traits by UBR5 also contribute to OC tumorigenesis. Hence, UBR5 exerts pro-tumorigenic function via both immunoregulatory and cell-autonomous mechanisms. This study expands the evolving understanding of the complex mechanisms by which E3 ligase dysregulation drives tumorigenesis.

Immune checkpoint blockade has elicited remarkable responses and gained regulatory approval for the treatment of multiple tumor types. Nevertheless, these strategies have shown limited efficacy in OC therapy. Their relative ineffectiveness has been

attributed to the lack of predictive biomarkers, low mutational burdens (except for BRCA1-associated OC), and the profoundly immunosuppressive TME[48]. Targeting UBR5 blunts tumor-induced immunosuppression and improves the response to anti-PD-1 antibodies by preventing intratumoral infiltration of TAMs, one of the most important players in shaping the TME in OC. Furthermore, although CAR-T therapy has generated promising data in OC preclinical models[49], we show here that its benefit can be greatly augmented by UBR5 inhibition to achieve durable survival. These findings highlight the value of normalizing TME in adoptive T cell therapies.

In summary, our findings reveal the oncogene-like properties of a novel molecule that has strong TME-modulating activities, particularly in TAM recruitment and maintaining cancer stemness. We demonstrate the therapeutic potential of targeting UBR5 to augment conventional treatment and substantially improve the outcome of immunotherapy. Since UBR5 dysregulation is a major risk factor also for breast and prostate cancers and others, the lessons learned here may have broader implications for improving the effectiveness of cancer therapy.

## Methods

**Cell culture**. Murine ovarian cancer cells ID8-Muc16^ecto^ were maintained in DMEM supplemented with 10% heat-inactivated fetal calf serum (FBS), 100 U/ml penicillin and streptomycin (P/S), and 2mM L-glutamine. Human ovarian cancer cells SKOV3 (MUC-16^ecto^/GFP-FFLuc) were cultured in RPMI 1640 supplemented with 10% FBS, nonessential amino acids, HEPES (N-2-hydroxyethylpiperazin-N'-2-ethanesulfonic acid) buffer, pyruvate, L-glutamine, penicillin/streptomycin, and 2-Mercaptoethanol. To establish stable UBR5 knockout cell lines, ID8-Muc16^ecto^ and SKOV3 (MUC-16^ecto^/GFP-FFLuc) were transfected with UBR5 CRISPR/Cas9 KO plasmid (Santa Cruz) and UBR5 HDR plasmid (Santa Cruz) using lipofectamine 3000 as per the manufacturer's protocol and selected with puromycin antibiotic. Stable UBR5 knockout monoclones were picked and verified after selection and propagation. To generate reintroduced cell lines in ID8/$Ubr5^{-/-}$ or ID8/GFP, cells were transfected with pCMV-Tag2B EDD, pCMV-Tag2B EDD C2768A (Addgene, #37188 and #37189, respectively), pEF1α-mCcl2/Csf-1, pCMV3-mCtnnb1 (Sino Biological) using lipofectamine 3000. The stable cell lines were selected with G418 or hygromycin B and confirmed by q-PCR and western blot. ID8 and SKOV3 cells were validated by karyotyping and all cells were routinely checked for mycoplasma contamination.

Human ovarian surface epithelial cells (HOSEpiC) were purchased from ScienCell Research Laboratories. Cells were cultured with Ovarian Epithelial Cell Medium (OEpiCM, Cat.#7311) in poly-L-lysine-coated vessels.

Primary mouse TAMs were isolated from the peritoneal cavity of ID8 tumor bearing mice (C57BL/6 background) on day 35 after tumor implantation and culture with DMEM based medium.

For T cell proliferation suppression assay, T cells isolated by Pan T Cell Isolation Kit ll (MACS) were labeled with CFSE, stimulated with anti-CD3 (1 μg/ml, eBioscience) and anti-CD28 (1 μg/ml, eBioscience), and cocultured with TAMs isolated from ID8 bearing mice at 3:1 ratio in 96-well flat-bottom plates. Seventy-two hours later, cells were analyzed by flow cytomertry.

**Mice and ovarian cancer models**. Female C57BL/6, Rag2$^{-/-}$ (B6 (Cg)-Rag2$^{tm1.1Cgn}$/J), CD8$^{-/-}$ (B6.129S2-Cd8$^{atm1Mak}$/J), CD4$^{-/-}$ (B6.129S2-Cd4$^{atm1-Mak}$/J) mice age 6–8 weeks were purchased from Jackson Laboratory. Female SCID-Beige mice were purchased from Taconic Biosciences. Mice were maintained in a light/dark cycle with free access to food and water, room temperature (25 °C), in 40–60% of humidity. All animal experiments were performed in accordance with National Institutes of Health guidelines for housing and care of laboratory animals after protocol (protocol Number 0701-569A) approved by IACUC at Weill Cornell Medicine.

For syngeneic intraovarian model, 6–8-week-old female C57BL/6 mice were anesthetized with isoflurane and a single dorsal incision was made to access to ovary. ID8 cells ($1.0 \times 10^6$) were then injected into the left ovarian bursa. Tumor-bearing mice were weighed twice a week, and tumors were surgically harvested at 8 weeks post-tumor implantation. Tumor size were captured with digital calipers and tumor volumes were calculated using the equation $(L \times W^2)/2$, where "L" = length and "W" = width.

For i.p. model, $1.0 \times 10^7$ ID8-Muc16^ecto^ and SKOV3 tumor cells were injected intraperitoneally and mice were euthanized at indicated time for analysis or monitored for survival. Nanoparticle treatments started 7 days post-tumor implantation, and injections were administered twice a week for 3 weeks. For lung metastasis model of ID8 cells, $1 \times 10^7$ ID8 cells were intravenously infused and mice were sacrificed on day 30 or day 60 for analysis.

For chemodrug treatment, 5 mg/kg of cisplatin (Biovision) was i.p. injected into ID8 bearing mice weekly from day 14, for a total of 4 doses. For antibody

treatment, ID8 tumor mice were given 250 μg/mouse of αPD-L1 (10F.9G2, BioXcell) or αPD-1 (PMP1-14, BioXcell) i.p. on day 7, 10, and 14 post tumor implantation, for a total of three doses.

Whole blood samples from three mice per group were collected in BD Microgard™ capillary blood collector with retro-orbital bleeding at indicated time to perform hematology and liver chemistry testing at Weill Cornell Medicine, the Laboratory of Comparative Pathology (LCP), NY, USA.

For CAR-T treatment, ID8 tumor-bearing mice were treated with 4H11-28z/IL-12 CAR T cells or CD19-targeted CAR T cells on day 14.

**Bioluminescent imaging**. SCID/Beige mice i.p. inoculated with SKOV3 were anesthetized and injected with 1.5 mg of D-luciferin at day 21. Animal were imaged using Xenogen IVIS imaging system with Living Image software (Xenogen Biosciences, Cranbury, NJ, USA).

**Patients and specimens**. Epithelial ovarian cancer (EOC) samples from individuals were obtained from patients at the Comprehensive Cancer Center of Drum Tower Hospital, Medical School of Nanjing University, China (Supplementary Table 3) who underwent cytoreductive surgery between 2015 and 2018. All patients gave written informed consent for the use of their specimens for medical research. All clinical samples were anonymously coded, and the protocol was approved by the committee for Ethical Review Board at the Comprehensive Cancer Center of Drum Tower Hospital (Nanjing, China).

**Nanoparticle synthesis**. Porous silicon-based multistage vector (MSV) nanoparticles were fabricated by electrochemical etching of silicon wafer, surface modified with 3-aminopropyltriethoxysilane (APTES), and conjugated with E-selectin thioaptamer (ESTA) as previously described[50]. To prepare Ubr5 siRNA polyplex, Ubr5 siRNA oligos (#1GCUUCUAAGUUAGAACACA; #2 GCAAA-TAGCATAAGAGCAA, Sigma) were mixed with PEG(5k)2–PEI(10k) (PEG–PEI) (nitrogen in cationic polymer: phosphorus in siRNA oligo ratio = 15:1) in 10 mM HEPES buffer containing 5% glucose, and incubated at 20 °C for 15 min. To load Ubr5 siRNA polyplex into ESTA-MSV, 200 μl polyplex suspensions containing 20 μg siRNA was mixed with $1 \times 10^9$ dry ESTA-MSV particles, and sonicated for 3 min on ice.

**RNA isolation, qRT-PCR, and RNA sequencing**. Total RNAs were extracted with the RNeasy Plus Mini Kit (QIAGEN, 74134), and cDNA was synthesized using the High Capacity cDNA Reverse Transcription Kit (Applied Biosystem, 4368814). Quantitative PCR was performed on QuantStudio™ 6 Flex System (Applied Biosystem) using PowerUp SYBR Green Master Mix (Thermo Fisher). The expression levels of target genes were normalized with Gapdh abundance. Primers used for RT-PCR are listed in Supplementary Table 5. Second generation of RNA sequencing was performed by GenoIMCs Core Facilities at Weill Cornell Medicine and differential gene expression analysis was performed using DEseq2 package.

**Flow cytometry**. Malignant peritoneal wash cells, isolated TAMs and cells from lung were stained with the following antibodies (listed in Supplementary Table 4) from Biolegend: CD3 (17A2), CD8α(53-6.7), NK1.1(PK136), CD11b (M1/70), F4/80(BM8), Gr-1(RB6-8C5), CD11c(N418), MHCll (M5/114.15.2), PD-L1 (B7-H1), Ly6C(HK1.4), Ly6G(1A8), CD170 (Siglec F, S17007L); antibodies from eBioscience: CD45 (30-F11), CD4(GK1.5), CD25(PC61.5), Foxp3 (FJK-16s), Arginase-1 (A1exF5), and APC-conjugated anti-MUC16 (Memorial Sloan Kettering Cancer Center monoclonal antibody facility).

Briefly, cells were incubated with Zombie UV Fixable viability dye (Biolegend) for 20 min at room temperature, followed by staining with appropriate surface antibodies in BD FACS Buffer in the presence of CD16/32 Fc block (BD clone 2.4G2). Intracellular staining was performed according to Foxp3/Transcription Factor Staining Buffer Set (eBioscience). Data acquisition was performed on FACSCalibur (BC Biosciences) and analyzed via FlowJo. To determine absolute cell counts of TAMs, eosinophils, monocytes, MDSCs, and DCs in peritoneal cavity, peritoneal cells were retrieved from ID8 tumor bearing mice and divided into equal parts after red blood cell lysing for subsequent FACS analysis. The total number of immune subsets in each part was evaluated by flow cytometer with phenotypic gating criteria and back-calculated with starting dividing numbers of peritoneal cells.

Tumor associated macrophages were sorted from peritoneal washes of OC bearing mice with Becton–Dickinson Influx.

**Western blot**. Cells were lysed in RIPA buffer (Thermal Scientific) and the lysates were centrifuged at $3000 \times g$ for 30 min at 4 °C. Supernatants were collected and protein concentration was quantified by Bio-rad protein assay (Bio-rad, 5000006). Cytoplasmic and nuclear extracts were separated from ID8 cells with NE-PER™ Nuclear and Cytoplasmic Extraction Reagents (Thermo Scientific, 78833). Cell lysates were subjected to SDS-PAGE and transfected to the PVDF membrane, followed by immunoblotting with specific antibodies listed in Supplementary Table 4.

**Immunostaining**. For immunofluorescent staining, cell samples were fixed with 3.7% paraformaldehyde (PFA) for 10 min and permeabilized with 0.5% Trition-X100 (in PBS) for 10 min at room temperature, followed by blocking and staining. IHC staining for tissue sections were performed as previously described[15]. Antibodies used for immunostaining are listed in Supplementary Table 4. Confocal microscopy images were taken under Zeiss 880 Laser Scanning Confocal Microscope and evaluated with ZEN2 software. Slides were observed and images were captured with Olympus BX60 Upright Microscope. Apoptosis in lung sections was measured by DeadEnd™ Colorimetric TUNEL System (Promega, G7360).

IHC analysis was performed by two independent observers who were blind to the results of other makers and the clinical outcome. Each section was randomly selected for five fields at low-power field (×100). For semiquantitative analysis of UBR5, the multiple of positive cell percentage score and intensity score was determined as previously described. The percentage of positive cells (PP) scored from 0 to 4 (0, none; 1, <10%; 2, 10–50%; 3, 51–80%; 4, >80%), and the staining intensity scored (SI) from 0 to 3 (0, none; 1, weak; 2, moderate; 3, strong). The semiquantitative immunoreactive score (IRS) = PP × SI. For evaluating the density of intratumoral CD68$^+$ cells, the respective areas were captured at ×200 magnification. The number of CD68$^+$ cells in each region was then counted.

**Spheroid cell proliferation assay**. The morphological and quantitative analysis of spheroid cell proliferation/ viability was determined with Cultrex® 3-D Spheroid Colorimetric Proliferation/Viability Assay kit (Trevigen). ID8 cells were seed at 3000/well and SKOV3 were seeded at 4000/well. Spheroids were photographed on day 5, and images were analyzed using ImageJ. At the endpoint (day 6), cell viability was assessed by absorbance at 570 nm using MTT.

**3D coculture of TAMs and ID8 cells**. 3D coculture spheroid formation assay was performed as previously described[23]. Briefly, mouse F4/80$^+$CD11b$^+$ TAMs isolated from OC-bearing mice and ID8 tumor cells were mixed at a ratio of 1:10 with a fixed total number of 40,000 cells/well in media containing 2% Matrigel and seeded onto Matrigel-precoated 24-well plate. The cell mixtures were incubated at 37 °C for 48 h and the number (per well) and size (area) of spheroids were quantified. Images were captured with Invitrogen™ EVOS™ FL Digital Inverted Fluorescence Microscope and analyzed using ImageJ software.

**ELISA**. A total of $4 \times 10^5$ ID8 cells were seeded in 6-well plate in DMED without FBS and cultured for 24 h before harvesting. ELISA on mCCL2 (DY479, R&D) and CSF-1(MMC00, R&D) were performed as instructed by manufacturer.

**Cell proliferation assay**. Cell proliferation was evaluated by Sulforhodamine B (SRB) assay. Briefly, cells ($1 \times 10^4$/well) were seeded into 96-well plates and cultured for 24, 48, and 72 h, then fixed with 10% trichloroacetic acid (Sigma, T8657) for 30 min at 4 °C and stained with 0.4% (w/v) SRB (Sigma, 230162) in 1% acetic acid solution for 30 min. After washing with 1% acetic acid, bound SRB was solubilized with 10 mmol/l Tris buffer, and absorbance (OD) was measured at 510 nm using 96-well plate reader.

**Cell invasion assay**. Cell invasion was assayed using Matrigel-coated BioCoat Cell Culture Inserts from Corning. $2 \times 10^4$ tumor cells starved overnight in serum-free medium were plated into each insert and 0.5 ml medium with 10% FBS were added into the bottom of each well. Non-invading cells were removed after 24 h of incubation. Transwell membrane was fixed with 4% paraformaldehyde and stained with 0.5% crystal violet. To count the fixed cells, five random fields of vision were captured and counted using phase contrast microscope. Independent experiments were performed in triplicate.

**Clonogenic assay**. Single cell suspension was seeded in triplicate 6-well culture plates with 100 ID8-Muc16$^{ecto}$ cells/well, 200 SKOV3 cells/well, and 200 OVCAR3 cells/well. Cells were cultured for 1–2 weeks to form colonies and fixed with methanol and then stained with 0.5% crystal violet. The number of clones formed in each well was counted and photographed. All assays were conducted in triplicate.

**Statistics and reproducibility**. Statistical analysis was performed using GraphPad Prism 8.4.3 software. In all figures, the data points and bar graphs represent the mean of independent biological replicates. Results are presented as mean ± SD or mean ± SEM. In all graphs, the error bars represent the standard deviation and are only shown for experiments with $n = 3$ or greater as indicated. All data are representative of at least two independent experiments with similar results. Comparisons between two groups were performed using an unpaired two-sided Student's $t$-test; comparison of multiple conditions was done with One-way ANOVA test. The relationships between UBR5 expression and CD68$^+$ macrophage infiltration or UBR5 expression, and the density of Ki67 in OC patient samples were analyzed using Pearson correlation coefficient. The differences in demographic characteristics of OC patients were analyzed by the $\chi^2$ test. $P < 0.05$ was considered statistically significant.

**Reporting summary**. Further information on research design is available in the Nature Research Reporting Summary linked to this article.

## Data availability

The RNA seq data generated in this study from murine ovarian tumor-associated macrophages have been deposited in NCBI GEO database under accession code GSE158911 . For the relationship between *TP53* mutation and *UBR5* gene alterations in serous ovarian carcinoma cohorts, the data in Fig. 5j, k were generated from publicly available datasets from the TCGA [https://www.cbioportal.org/study/summary?id=ov_tcga]. The authors declare that all data supporting the findings of this study are available within the article and Supplementary information files and from the corresponding author upon reasonable request. A reporting summary for this article is available as a Supplementary Information file. Source data are provided with this paper.

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

## Acknowledgements
This work was supported in part by an NIH award (R03CA230573 to X.M.); by an award from the Emerson Collective Cancer Research Fund (ECCRF-COR08 to X.M.); by an NIH award (5 P01 CA190174 to R.J.B.); by an institutional award (MSKCC-GC238051 to R.J.B.); by NIH awards (R01CA222959 and R01CA193880) to H.S.

## Author contributions
M.S. designed and performed the majority of the experiments and wrote the manuscript; O.Y., S.R., and T.P. designed and performed all CAR-T experiments; X.D. prepared lung cryosections and contributed to establish the syngeneic intraovarian model; L.Z. did human OC tissue preparation and analysis; T.Z. carried out the RNA-seq analyses; H.W. analyzed the relationship between *Tp53* and *UBR5* in a TCGA cohort; Z.Y. contributed to examining β-catenin expression and data analysis; J.M. and H.S. did the nanoparticle design and preparation; B.N. and M.L. contributed to some of the TAM analysis by FACS; R.B. and X.M. were responsible for the overall supervision of the project and funding as well as the writing of the manuscript.

## Competing interests
The authors declare no competing interests.
