## [Peer Review File · Nature Communications]

Reviewers' comments:

Reviewer #1 (Remarks to the Author):

Review_ “Tumor derived UBR5 promotes ovarian cancer growth and metastasis through inducing immunosuppressive macrophages and spheroids”

Manuscript ID: NCOMMS-20-04534

Song M. et al

Functional mechanisms of the E3 Ubiquitin Ligase UBR5 (EDD1) in cancer development and the clinical possibility for applying UBR5-based targeted therapies have been widely studied. The paper by Song M. et al. reported a novel mechanism that tumor-derived UBR5 promotes the peritoneal metastasis of OC through regulating adaptive immuneresponses such as recruiting TAMs and cell-autonomous mechanisms. The concept of this study is very important, particularly in identifying the possibility of developing new therapeutic strategies against ovarian cancer through UBR5. While this concept is not extremely novel, given reports on UBR5 driving metastasis of TNBC has been published by the same group at 2017, and the similarity on pathological characteristics between TNBC and HGSC. The experimental design in manuscript is well executed and contains an impressive set of data to analyze effects and mechanism of UBR5 in transforming immunomodulation of TME components as well as cancer stem cells. Overall, the authors presented a profound analysis of regulatory mechanism for UBR5 in regulating cell proliferation, migration, immunoadaptive responses through assays using cell-based, 3-D spheroids, and animal models.

However, I have some major concerns on this manuscript. First, the in vivo evidence by using i.v. & i.p. ID8 models and additional SKOV3 (i.p.) presented in this manuscript was not compelling for studying the hematogenous metastasis in ovarian cancer, for the reason that trace of ovarian cancer metastasis started from early spread into the abdomen, and followed with diffuse peritoneal invasion, omental caking and ascites. To reach authors' fundamental goal in investigating the effects of targeting tumor-driven UBR5 in metastatic spreading of ovarian cancer, an intra-ovarian model of syngeneic or orthotopic high-grade OC model surgically implanted into ovary is much more preferential. Based on the standard requirement at the Nature publication group, an accurate in vivo model to address the core question is essential.

Second, the clinical application for targeting UBR5 or other E3 ligase inhibitor is facing the same question, its broad-substrate spectrum and tight regulating as transcription factors. Therefore, the toxicity for this kind of strategy is among the biggest hindrances during clinical development. On this

ground, I would like to see the toxic validation on the in vivo experiments presented in this study. Unfortunately, the authors did not present any kinds of these data among all of the animal models in this manuscript, from as simple as body weight to the pathological index at the end of expts. In particular, the reading of CBC, LFT and other organs are important in the ID8/Ubr5^{-/-} mice with or w/o other additional treatments, because UBR5 is high expressed in lymphocytes based on GTex portal. For the same reason, for the in vitro function assays targeting UBR5 in migrating, proliferating, I would recommend author to include a “normal ovarian epithelial cell (it can be immortalized)” as control to ID8 or SKOV3.

Third, I have concerns on the OC patient cohort that was used for UBR5 and CD68 staining and correlative studies presented at the results description for Figure 7. There was no description of ovarian patients in the material and method (50 patients for Fig 7a, or 26 patients for Fig7b or 7C—I don't know why there were differences between the panels since it appeared that author used the same cohort), if they have received the similar adjuvant chemotherapy (i.e. cisplatin or combinational therapies with other adjuvants), and if, any of the regimes with UBR5 expression were associated with an enhanced survival effect; hence, authors failed to present the evidence showing the OC patient cohorts may not have sufficient power to perform these analyses. Also, the approval of the related IRB protocol should be mentioned in the M&M.

There are few concerns other than the issues listed above:

Fig 1 Measuring the ascitic fluid volume and the i.p. injected syngeneic ID8 numbers to support the statement on Page 7 “ID8/Ubr5^{-/-} failed to progress from lung micrometastases to macroscopic metastases” was not accurate, unless author can present the data showed that ID8/Ubr5^{-/-} cells did have the same take rate ability as ID8 wt. In addition, the primary OC mets should be measured at liver, mesentery in addition to lung.

Supplemental Figure 3i. The statistic differences between ID8GFP versus ID8-Ubr5^{-/-} on monocytes, macrophages, DC, NK cells appeared to be very similar at least. Could author clarify the calculation and statistic method used in here?

The labeling in Fig 5f might be off. Could author clarify what is conclusion coming out from comparing ID8/UBR5^{-/-}+siR-ctl with ID8/GFP+siR-Tp53?

I would ask author to please unify the usages of abbreviation, such as UBR5 and EDD. Because it is quite confusing, especially for readers not in the E3 ligase field.

Overall, this manuscript needs stronger data sets (both in vivo and in vitro) to support their conclusion. By all means, if authors can address above comments, I am happy to review it again.

Reviewer #2 (Remarks to the Author):

In this manuscript the authors assess the consequences of knock-out of Ubr5 in cell line and mouse models of ovarian cancer. The bulk of the data is from a single Ubr5-KO clone of ID8, a commonly used mouse cell line of ovarian cancer. Some confirmatory testing is performed using a second UBR5-KO clone and the human OC cell line SKOV3 with knock-out or over-expression of UBR5.

Ubr5-KO resulted in slowed tumor growth, increased survival, mesenchymal morphology, reduced numbers of tumor associated macrophages (TAMs) and an M1 TAM phenotype as opposed to M2 for parental ID8 tumors. In human OC tumors a positive association between the level of UBR5 and the M2 marker CD68 was identified.

P53 levels were increased in Ubr5-KO tumors and knockdown of p53 in these tumors restored B-catenin levels to control levels, leading to the conclusion that UBR5 regulates B-catenin via p53 pathways.

Over-expression of UBR5 increased resistance to cisplatin and its loss increased response to treatment with anti-PD1 and CAR T-cells targeting MUC16.

This is a very thorough study of the consequences of Ubr5 loss in ovarian cancer. While the study was limited by most of the work being performed in a single cell line, confirmatory support in another cell line was provided in some cases.

The results will be of interest to the ovarian cancer community. It adds new mechanistic insights into UBR5 which has previously been reported as a negative prognostic factor and attractive therapeutic target for ovarian cancer (ref 16,17). It confirms previous reports that over-expression of UBR5 increases resistance to cisplatin.

A previous report investigating the outcomes of EDD/UBR5 knockdown (using siRNA; Ref 16) found that EDD knockdown rapidly reduced cell numbers and this was due to the rapid induction of apoptosis. No markers of apoptosis have been assessed in this study. Please explain the results in the context of the previously published results and how apoptosis was determined not to contribute to the increased survival etc in Ubr5-KO models presented in the current study.

In the context of ovarian cancer, sub-type relevance needs to be considered. This is critical as different sub-types are essentially different diseases with activation of different pathways and different responses to treatment and it is important to specify the sub-type to which the findings have relevance. The most common and aggressive ovarian cancer is high grade serous ovarian cancer. It accounts for ~ 80% of epithelial ovarian cancers, patients are often initially sensitive to cisplatin and loss or dysfunction of p53 due to mutation of Tp53 is virtually pathognomic for this subtype. As such the identified UBR5 regulation of p53 by UBR5 may not occur in high grade serous ovarian cancer. S Fig 1 shows data from TCGA, some of which has referenced in the Introduction and previously published. These data should be divided by OC-subtype and by TP53 mutation and expression to identify the sub-type(s) most relevant to this study. What is the relationship between p53 and UBR5 in TCGA data? What is the relevance of UBR5 regulation of p53 when Tp53 is mutated?

Sub-type classification, grade and stage for the tumors assessed in Fig 7 is also important to understanding the sub-type relevance.

My other comments refer primarily to the interpretation of the results which are sometimes overstated and in many cases is converted to UBR5 overexpression whereas the results relate to knock-down. This may not necessarily follow and would need to be conclusively shown. Some examples follow. However, this is a major issue for the entire manuscript including the title, and changes in addition to those below need to be made.

P6 Results - States that UBR5 amplification shortened survival. The data shows it has a significant association, but this does not prove cause. This statement needs to be rewritten.

P8 Results - States that results validate tumor-promoting effects of UBR5. However, a positive relationship was not demonstrated, rather the results show that Ubr5 loss slowed tumor growth and spread. This statement needs to be rewritten.

P9 Results - States that UBR5 plays a critical role in regulating macrophage recruitment into ID8 tumors – a positive relationship was not demonstrated, rather a negative relationship with a reduction of TAMs associated with Ubr5 KO. This statement needs to be rewritten.

P13 – States that RNA-seq was performed on primary ID8 tumors and then states results for peritoneal TME. This is inconsistent. Fig 6 legend states that RNA seq was performed on recovered peritoneal cells from ID8 bearing mice. Which is correct? Was the TME separated from the primary tumour cells. Please clarify and be consistent. This also needs to be changed / clarified in the Discussion P17.

P15 Results states that tumor-derived UBR5 may drive human OC progression. This was not shown, rather, results show that knock out of UBR5 increased survival and decreased Ki67 etc.

Discussion: Again, the study did not reveal UBR5 as a pivotal drive of OC aggressiveness. The bulk of the data showed that loss of Ubr5 resulted in a less aggressive phenotype of OC.

“We find that tumor-derived UBR5 promotes OC peritoneal implantation...” The bulk of the results again were related to what happened with UBR5 loss.

Other comments

Fig 5d and Results p12 - What is meant by UBR5-overexpression ID8/GFP tumor – is this the parental cell line with endogenous UBR5 or has Ubr5 been overexpressed? It appears to be the parental line – this needs to be clarified.

Figure 1 legend

d – why are these Muc16+ cells whereas elsewhere they are referred to as ID8 cells?

K - modality” should be “mortality”

I - “reconstituted” (here and throughout the manuscript) is not the correct word.

O – why was data normalized to PAC rather than GAPDH as stated in the Methods?

Figure 6 – do error bars denote SEM or SD?

SFig 5 – h – spelling mistake

Ethics protocol IDs and information is missing for both mouse and human studies.

Some grammar and language editing is required – eg “modality” should be “mortality” in figure legends (eg Fig 8, SFig 2)

Results page 8 – “T cell frequency” does not make sense and should possibly be “T cell infiltration” or simply T cell numbers”

Results page 14 second paragraph – “foregoing” does not make sense

Figure 7 - what is IRS

Discussion p18 – “annulling” is not the correct word and needs to be changed; UB55 should be UBR5; “Reconstitution” is not correct word, see below.

Methods p21 – “reconstituted” (here and throughout the manuscript) is not the correct word. Suggest using something like “re-expression of Ubr5” in Ubr5^{-/-} cells and “over-expression of Ubr5” in parental cell.

Page 22 Methods – “since day” should be “from day”

Methods for cell viability and sulforodamine assays appear to be missing.

Data and materials availability - this statement needs clarification as it does not make sense. Access to the data needs to comply with the journal requirements.

Reviewer #3 (Remarks to the Author):

Song M and his colleagues studied that tumor derived UBR5 is a pivotal driver of OC aggressiveness. Depletion of Ubr5/UBR5 in both mouse ID8-Muc16ecto OC and human SKOV3 OC blocked tumor growth and peritoneal metastasis by disrupting paracrine regulation of TAM infiltration and cell-intrinsic regulation of spheroid formation. Targeting Ubr5 in ID8 OC synergistically improved therapeutic outcomes of chemotherapy and immunotherapies with immune-checkpoint blockade and CAR T cell administration. This work has elucidated the causality and mechanisms of UBR5’s tumorigenic activities in OC and broadened our understanding of the biology of a novel E3 ligase in regulating cancer-immune cell crosstalk. Reducing the expression of UBR5 could be an efficacious

therapeutic approach for OC. Although there are several new aspects reported, experimental and interpretive issues require further investigation.

1. Author considered β -catenin as epithelial marker and claimed that loss of β -catenin in UBR5 deficient cells demonstrated epithelial to mesenchymal transition. On contrary, many other studies suggest activation of Wnt/ β -catenin signalling induces EMT - (1) it down-regulates E-cadherin expression via the transcription factors Twist and Slug, (2) it up-regulates adhesion molecules that favor cell motility, such as N-cadherin and L1, and (3) it induces proteases and other EMT promoters. Wnt signaling can therefore induce a cadherin switch and weaken cell–cell adhesion (PMID: 20182623, 31101875, 30271576).

Additionally, the author claimed that “It was noteworthy that knocking down p53 expression in ID8/Ubr5^{-/-} cells restored β -catenin to the control level (Fig.5g)”, which is in corroboration with the previous publication, which also claimed that loss of p53 restores the transcriptional activity of β -catenin which further induce EMT (PMID: 24023784). What is the status of EMT markers in ID8/Ubr5^{-/-} after the restoration of β -catenin (if the author claims β -catenin as an epithelial marker) by p53 knockdown?

2. According to results in figure 1, the author says UBR5-deficient cells show more EMT and undergo more metastasis in lung till day 30. But, because of impaired MET, UBR5-deficient cells do not form macro-metastasis in the lung. The finding should be confirmed by analysis of more EMT markers (like E-cadherin, Cytokeratin 18, ZO-1, Fibronectin 1, Vimentin, N-cadherin, Snail, Zeb1) on protein level in vitro results. Additionally, the author needs to perform IFC analysis (EMT markers+ID8+UBR5) in the lungs to claim EMT at day 30 and impaired MET at day 60 tumors, and to show ID8/Ubr5^{-/-} cells are survived in lung microenvironment after 60 days in circulation. Because in the discussion, the author claimed that “UBR5 has been shown to suppress death receptor-induced apoptosis and downregulate proapoptotic MOAP-1 in human OC cells. However, we observed that genes involved in apoptotic pathways were only slightly altered by Ubr5 deficiency at the mRNA level (data not shown).”

3. In the whole manuscript, the β -catenin level was checked by using whole cell extract, which may indicate the cytoplasmic concentration of β -catenin (transcriptionally inactive). At least critical experiments in the manuscript (fig.1f, fig. 5g, fig 6c) need to show the transcriptional activity of β -catenin either by WB of β -catenin in nuclear and cytoplasmic fraction or TCF/LEF activity assay along with protein expression of Wnt target genes like CCND1, MYC; etc.

4. What is the nodule count at day 30 in figure 1e?

5. The heat map in supp. fig 5a need to show the name of genes are in G1 and G2?

6. The author demonstrated that “TAMs from Ubr5^{-/-} tumor-bearing mice exhibited higher levels of M1 macrophage markers (e.g. il12a, Ccr2) and lower levels of M2 markers (e.g. Cx3cr1, il10) (Fig.4b and Supplementary Fig.5b).” Change in macrophage phenotype in the tumor microenvironment does affect the phenotype of T cells, so what is the explanation behind not observing the significant shift in the infiltration or phenotype of any other immune cells?

7. According to figure 6a, Ccl2, Vegf, Il6, Csf-1, Cxcl1, Genes in Wnt signaling were significantly down-regulated in ID8/Ubr5^{-/-} bearing mice (Fig.6a). Therefore, the author reintroduces CCL2, M-CSF, and β -catenin in ID8/Ubr5^{-/-} cells. But according to figure 6c, the reintroduction of CCL2, M-CSF, and β -catenin do not restore protein expression of UBR5 in ID8/Ubr5^{-/-} cells (pink block). In contrast, all UBR5-related functional effects (Fig.6 d-f and Supplementary Fig.6 e-i) were restored. Is it means UBR5-related functional results are independent of the protein level of UBR5?

8. Figure 6 showed the combined effect of the reintroduction of CCL2, M-CSF, and β -catenin in ID8/Ubr5^{-/-} cells. What are the results, if CCL2, M-CSF, and β -catenin reintroduce separately in ID8/Ubr5^{-/-} cells?

9. In the discussion, author claimed that “Given that UBR5 didn’t modulate their mRNA stability (Supplementary Fig.6c), UBR5 may stabilize transcription factors (TFs) that control Ccl2/Csf1 gene transactivation or act as coactivator of these TFs”. In the present study, the author also showed modulation of β -catenin (which is also a TF) in UBR5-deficient cells. It is important to check the role of UBR5-dependent β -catenin regulation in Ccl2/Csf1 gene transactivation because β -catenin is known to play a role in transactivation of CCL2 (PMID: 16003740).

10. The previous publication from the author (E3 Ubiquitin Ligase UBR5 Drives the Growth and Metastasis of Triple-Negative Breast Cancer) showed similar findings in breast cancer. Why the author did not explore UBR5 dependent molecular and functional role in the modulation of TAMs (like fig 2), cytokines, chemokines, and β -catenin (like fig. 5 and 6), Chemotherapeutic and Immuno-Therapeutics therapeutic advantages (like fig. 7 and 8) in breast cancer. What is the rationale behind performing follow up study in ovarian cancer and not in breast cancer? Are the molecular events demonstrated in the present study are only observed in ovarian cancer?

Point to Point Response

We are very grateful of all three reviewers' unanimous recognition of the novelty and importance of our work and thankful to their highly constructive and detailed critiques, which have prompted us to make an all-out effort to address all issues raised, major or minor, for the improvement of the manuscript to the best of our ability and to the standards of the journal. As you will see from the "point-to-point-response" below, we have left no stone unturned in addressing the reviewers' concerns in a complete and rigorous manner.

Reviewer #1:

Q1: First, the in vivo evidence by using i.v. & i.p. ID8 models and additional SKOV3 (i.p.) presented in this manuscript was not compelling for studying the hematogenous metastasis in ovarian cancer, for the reason that trace of ovarian cancer metastasis started from early spread into the abdomen, and followed with diffuse peritoneal invasion, omental caking and ascites. To reach authors' fundamental goal in investigating the effects of targeting tumor-driven UBR5 in metastatic spreading of ovarian cancer, an intra-ovarian model of syngeneic or orthotopic high-grade OC model surgically implanted into ovary is much more preferential. Based on the standard requirement at the Nature publication group, an accurate in vivo model to address the core question is essential.

A1: We fully appreciate the reviewer's suggestion. We have now used an intra-ovarian mouse model in which syngeneic epithelial ovarian cancer cells (ID8) were seeded into the ovarian bursa and compared the progression of ID8/GFP vs ID8/*Ubr5*^{-/-} tumors. Similar to the *ip* model, ID8/*Ubr5*^{-/-} bearing mice displayed strongly reduced tumor burden, impaired ascites accumulation, diminished tissue implantation and attenuated tumor-induced splenomegaly, with prolonged survival, compared with ID8/GFP bearing mice. These new data are included and highlighted in Fig.1f-h and Supplementary Fig.3a-e.

The body weight of tumor bearing mice was measured and expressed as percent weight gain (shown in Supplementary Fig.3e). Blood samples were collected 7 weeks after tumor inoculation for CBC/LFT tests, and CBC/LFT parameters are shown in Supplementary Table1. Values of hematological and chemical blood parameters in both types of tumor bearing mice were within the normal range.

The surgical procedures for intrabursal injection of tumor cells are incorporated in *Methods* and highlighted.

*Q2: Second, the clinical application for targeting UBR5 or other E3 ligase inhibitor is facing the same question, its broad-substrate spectrum and tight regulating as transcription factors. Therefore, the toxicity for this kind of strategy is among the biggest hindrances during clinical development. On this ground, I would like to see the toxic validation on the in vivo experiments presented in this study. Unfortunately, the authors did not present any kinds of these data among all of the animal models in this manuscript, from as simple as body weight to the pathological index at the end of expts. In particular, the reading of CBC, LFT and other organs are important in the ID8/*Ubr5*^{-/-} mice with or w/o other additional treatments, because UBR5 is high expressed in lymphocytes based on GTex portal. For the same reason, for the in vitro function assays targeting UBR5 in migrating, proliferating, I would recommend author to include a "normal ovarian epithelial cell (it can be immortalized)" as control to ID8 or SKOV3.*

A2: Per the request of the reviewer, we monitored body weight of tumor bearing mice (ID8/GFP vs. ID8/*Ubr5*^{-/-} vs. ID8/GFP+hUBR5) with and w/o cisplatin treatment. Blood samples were collected 5 weeks after tumor inoculation (1 week following discontinuation of cisplatin treatment) for CBC/LFT tests. No significant differences were detected for all CBC/LFT parameters between ID8/GFP and ID8/*Ubr5*^{-/-} groups, and all parameters were within the normal ranges. However, ID8/GFP+hUBR5 bearing mice showed decreases in red blood cell, hemoglobin, hematocrit and reticulocyte counts compared to either ID8/GFP or ID8/*Ubr5*^{-/-} bearing mice, most likely as a consequence of a more rapid tumor progression. The growth of ID8/GFP+hUBR5 tumors also caused decreases in the levels of alkaline phosphate (ALP) and albumin

(ALB). After cisplatin treatment, ID8/*Ubr5*^{-/-} bearing mice displayed a slight decrease in lymphocyte ratio and ALP level, while increased alanine transaminase (ALT) in blood, compared to the other two groups. But all hematological and chemical blood parameters were within the normal range. These data demonstrate that UBR5-deficient tumors did not cause any significant alterations in the blood, nor pronounced pathogenesis in mice as a whole. Additional cisplatin treatment did not result in greater toxicities in UBR5-deficient tumor-carrying mice than expected of the drug on its own.

CBC/LFT parameters in these tumor bearing mice are shown in Supplementary Table2. Body weight gains are shown in Fig 8d. For *in vitro* functional assays, we used a human ovarian surface epithelia cell line (HOSEpiC) as control to compare protein expression, cell proliferation, migration and spheroid formation in human ovarian cancer cell lines (SKOV3 and OVCAR3) with *Ubr5* depletion. The data are included in Fig.7d-e and Supplementary Fig.9c-j. The HOSEpiC culture method is added in *Methods* and highlighted.

Q3: *Third, I have concerns on the OC patient cohort that was used for UBR5 and CD68 staining and correlative studies presented at the results description for Figure 7. There was no description of ovarian patients in the material and method (50 patients for Fig 7a, or 26 patients for Fig7b or 7C—I don't know why there were differences between the panels since it appeared that author used the same cohort), if they have received the similar adjuvant chemotherapy (i.e. cisplatin or combinational therapies with other adjuvants), and if, any of the regimes with UBR5 expression were associated with an enhanced survival effect; hence, authors failed to present the evidence showing the OC patient cohorts may not have sufficient power to perform these analyses. Also, the approval of the related IRB protocol should be mentioned in the M&M.*

A3: We have now added the demographic characteristics of epithelial ovarian cancer (EOC) patients in Supplementary Table 3, and clinical specimen/IRB protocol information in *Methods*. We used the same EOC patient samples in Fig. 7a-c, and some of these specimens with equivalent CD68⁺ or Ki67⁺ cell density displayed the same levels of UBR5 expression. So there were overlapped individual dots in Fig7b and c with Pearson correlation analysis. The purpose of using these clinical specimens was to explore the potential correlation between UBR5 expression and macrophage infiltration/tumor cell proliferation, not the relationship between UBR5 expression and EOC survival/recurrence *et. al.*, which is described in Supplementary Fig1. Admittedly, more clinical specimens should be collected to address the relationship between UBR5 expression and therapeutic benefit of chemotherapy regimens, but this was not a major focus of the present work. We are continuing to collect human patient specimens for further studies.

Q4: *Fig 1 Measuring the ascitic fluid volume and the i.p. injected syngeneic ID8 numbers to support the statement on Page 7 “ID8/*Ubr5*^{-/-} failed to progress from lung micrometastases to macroscopic metastases” was not accurate, unless author can present the data showed that ID8/*Ubr5*^{-/-} cells did have the same takerate ability as ID8 wt.*

A4: We drew the conclusion that “ID8/*Ubr5*^{-/-} failed to progress from lung micrometastases to macroscopic metastases” based on our observations in an *i.v.* injection model of ID8, not an *i.p.* model. In this *i.v.* model, more ID8/*Ubr5*^{-/-} cells were detected in lung by FACS at 30 days post injection, but far fewer pulmonary metastatic nodules were seen in mice bearing ID8/*Ubr5*^{-/-} at 60 days post injection (Fig 1c-d, and Supplementary Fig. 2g,h).

Q5: *Supplemental Figure 3i. The statistic differences between ID8GFP versus ID8-*Ubr5*^{-/-} on monocytes, macrophages, DC, NK cells appeared to be very similar at least. Could author clarify the calculation and statistic method used in here?*

A5: We added the calculation method for absolute cell count of immune subsets in *Methods* section with

highlighting: “To determine absolute cell counts of TAMs, eosinophils, monocytes, MDSCs and DCs in peritoneal cavity, peritoneal cells were retrieved from ID8 tumor bearing mice and divided into equal parts after red blood cell lysing for subsequent FACS analysis. The total number of immune subsets in each part was evaluated by flow cytometer with phenotypic gating criteria and back-calculated with starting dividing numbers of peritoneal cells.”

Q6: *The labeling in Fig 5f might be off. Could author clarify what is conclusion coming out from comparing ID8/UBR5^{-/-}+siR-ctl with ID8/GFP+siR-Tp53?*

A6: The differences in spheroid formation between ID8/*Ubr5*^{-/-}+ siR-*Ctrl* and ID8/GFP+siR-*Tp53* are now presented in Fig. 5h (formerly Fig. 5f).

Q7: *I would ask author to please unify the usages of abbreviation, such as UBR5 and EDD. Because it is quite confusing, especially for readers not in the E3 ligase field.*

A7: We now use h*UBR5* as the abbreviation for human *UBR5* gene throughout the manuscript.

Reviewer #2:

Q1: *A previous report investigating the outcomes of EDD/UBR5 knockdown (using siRNA; Ref 16) found that EDD knockdown rapidly reduced cell numbers and this was due to the rapid induction of apoptosis. No markers of apoptosis have been assessed in this study. Please explain the results in the context of the previously published results and how apoptosis was determined not to contribute to the increased survival etc in Ubr5-KO models presented in the current study.*

A1: We are aware of that work. To directly address the reviewer’s question, we took both the *in vitro* and *in vivo* approaches. *In vitro*, we observed that *Ubr5* depletion in ID8 resulted in elevated PARP expression compared to WT ID8 cells but the relative amount of PARP cleavage, which is considered a hallmark of apoptosis, was not increased (Fig. 5d). *In vivo*, ID8/*Ubr5*^{-/-} tumor bearing mice did not display enhanced TUNEL-positive signals in the lungs with *i.v.* injected tumor cells, compared to the control group (Fig. 5e). Collectively, these data suggest that apoptosis is unlikely a major cause of impaired tumor growth of ID8/*Ubr5*^{-/-}.

Q2: *In the context of ovarian cancer, sub-type relevance needs to be considered. This is critical as different sub-types are essentially different diseases with activation of different pathways and different responses to treatment and it is important to specify the sub-type to which the findings have relevance. The most common and aggressive ovarian cancer is high grade serous ovarian cancer. It accounts for ~ 80% of epithelial ovarian cancers, patients are often initially sensitive to cisplatin and loss or dysfunction of p53 due to mutation of *Tp53* is virtually pathognomic for this subtype. As such the identified UBR5 regulation of p53 by UBR5 may not occur in high grade serous ovarian cancer. S Fig 1 shows data from TCGA, some of which has referenced in the Introduction and previously published. These data should be divided by OC-subtype and by TP53 mutation and expression to identify the sub-type(s) most relevant to this study. What is the relationship between p53 and UBR5 in TCGA data? What is the relevance of UBR5 regulation of p53 when *Tp53* is mutated? Sub-type classification, grade and stage for the tumors assessed in Fig 7 is also important to understanding the sub-type relevance.*

A2: (1) Per the request of the reviewer, we further delineated the relationship between *Tp53* and *UBR5* in a TCGA cohort of about 600 serous ovarian cancer (SOC) samples. Similar ratios of *UBR5* alterations among

TP53 wild type, mutated, and not profiled subgroups were observed (Fig5j). However, in a high-grade serous ovarian carcinoma (HGSOCs) cohort, we found a clear association between *UBR5* alterations and *TP53* mutations, i.e., cancers with *UBR5* gene alterations, predominantly amplification and gains, also harbored *TP53* mutations, whereas those in the diploid state (without *UBR5* alterations) were mostly without *TP53* mutations (Fig5k), suggesting a possible regulatory relationship between *UBR5* alteration and *TP53* mutation. (2) We also assessed the subtype/grade/stage classification of OC patient samples in Fig7 and the data are presented in Supplementary Table 3.

Q3: *My other comments refer primarily to the interpretation of the results which are sometimes overstated and in many cases is converted to UBR5 overexpression whereas the results relate to knock-down. This may not necessarily follow and would need to be conclusively shown. Some examples follow. However, this is a major issue for the entire manuscript including the title, and changes in addition to those below need to be made.*

P6 Results - *States that UBR5 amplification shortened survival. The data shows it has a significant association, but this does not prove cause. This statement needs to be rewritten.*

A3: The reviewer's point is well taken. We have changed this particular statement to "High expression of *UBR5* was associated with poorer patient prognosis and shortened survival rates". Regarding the title of the manuscript, we feel that we have used both knockout/knockdown as well as overexpression approaches to demonstrate the proactive role of *UBR5* in OC aggression in a cause-consequence manner. Specifically, we show that overexpression of *UBR5* in ID8/GFP further augmented tumor progression, with increased *CCL2* and *CSF-1* expression, TAM infiltration, cellular proliferation, spheroid accumulation, and shortened survival (Fig 6b-k). In Fig 8c and 8d, we show that overexpression of h*UBR5* in WT ID8 cells or reintroduction of h*UBR5* expression to *Ubr5*^{-/-} tumors made them more aggressive *in vivo* causing faster tumor growth and animal death. However, the same cannot be said about the role of spheroids. Thus, we have removed this word from the manuscript title.

Q4: *P8 Results - States that results validate tumor-promoting effects of UBR5. However, a positive relationship was not demonstrated, rather the results show that Ubr5 loss slowed tumor growth and spread. This statement needs to be rewritten.*

A4: We have changed the statement to "These data demonstrate a strong role of *UBR5* in ID8 tumor growth".

Q5: *P9 Results - States that UBR5 plays a critical role in regulating macrophage recruitment into ID8 tumors – a positive relationship was not demonstrated, rather a negative relationship with a reduction of TAMs associated with Ubr5 KO. This statement needs to be rewritten.*

A5: We have rewritten this statement as "Together, these data suggest that tumor derived *UBR5* plays a critical role in regulating macrophage recruitment into ID8 tumors." (Currently P10)

Q6: *P13 – States that RNA-seq was performed on primary ID8 tumors and then states results for peritoneal TME. This is inconsistent. Fig 6 legend states that RNA seq was performed on recovered peritoneal cells from ID8 bearing mice. Which is correct? Was the TME separated from the primary tumor cells. Please clarify and be consistent. This also needs to be changed / clarified in the Discussion P17.*

A6: The samples for RNA-seq were retrieved peritoneal cells from ascites of ID8 tumor bearing mice, which contained non-attached ID8 tumor cells and infiltrating immune subsets. Ascites constitute a unique OC tumor microenvironment (TME). We have changed the statement on P14 (originally P13) to: "we performed

RNA-seq analyses with retrieved peritoneal cells from ascites.....”

Q7: *P15 Results states that tumor-derived UBR5 may drive human OC progression. This was not shown, rather, results show that knock out of UBR5 increased survival and decreased Ki67 etc.*

A7: We have changed the statement to: “tumor-derived UBR5 is required for human OC progression” (currently P17).

Q8: *Discussion: Again, the study did not reveal UBR5 as a pivotal driver of OC aggressiveness. The bulk of the data showed that loss of Ubr5 resulted in a less aggressive phenotype of OC. “We find that tumor-derived UBR5 promotes OC peritoneal implantation...” The bulk of the results again were related to what happened with UBR5 loss.*

A8: We have shown that compared to ID8/GFP, overexpression of UBR5 in ID8/GFP further augmented tumor progression, with increased CCL2 and CSF-1 expression, TAM infiltration, cellular proliferation, spheroid accumulation, and shortened survival (Fig 6b-k). In addition, UBR5 overexpression largely abrogated the therapeutic effect of cisplatin treatment in tumor bearing mice (Fig 8b). Thus, we think the statement “UBR5 as a pivotal driver of OC aggressiveness” in *Discussion* is not inaccurate.

Q9: *Fig 5d and Results p12 - What is meant by UBR5-overexpression ID8/GFP tumor – is this the parental cell line with endogenous UBR5 or has Ubr5 been overexpressed? It appears to be the parental line – this needs to be clarified.*

A9: UBR5-overexpressing ID8/GFP is not parental cell line, but UBR5 overexpressed stable cell line by transfection of plasmid pCMV-Tag2B EDD into ID8/GFP. Detailed information for this cell line is presented in *Methods*.

Q10: *Figure 1 legend d – why are these Muc16+ cells whereas elsewhere they are referred to as ID8 cells? K - modality” should be “mortality” I - “reconstituted” (here and throughout the manuscript) is not the correct word. O – why was data normalized to PAC rather than GAPDH as stated in the Methods?*

A10: (1) The ID8 cells we used in this manuscript are expressing Muc16^{ecto}, and are simplified as ID8. We have clarified this in *Results* (P6). To avoid confusion, we have changed ID8-Muc16^{ecto} in Fig1 legend to ID8;

(2) Fig1k “modality” has been corrected to “mortality”;

(3) We have changed “reconstitution” to “reintroduction”;

(4) Fig1 o, we performed RT-PCR with peritoneal cells from ascites which contain ID8 tumor cells, infiltrating immune cells and stromal cells *et.al*. Both ID8/GFP and ID8/*Ubr5*^{-/-} cells contain puromycin N-acetyl- transferase (PAC), and could be used to distinguish tumor cells from other cells (such as infiltrating immune cells, stromal cells, endothelial cells *et.al*).

Q11: *Figure 6 – do error bars denote SEM or SD? SFig 5 – h – spelling mistake Ethics protocol IDs and information is missing for both mouse and human studies.*

A11: (1) Error bars denote SD in Fig 6, and we have clarified it in the figure legend. (2) Spelling mistake in Supplementary Fig 7c (originally Supplementary Fig 5h) has been corrected. (3) Ethics protocol IDs and information for mouse and human studies has been added and highlighted in *Methods*.

Q12: *Some grammar and language editing is required – eg “modality” should be “mortality” in figure*

legends (eg Fig 8, SFig 2) Results page 8 – “T cell frequency” does not make sense and should possibly be “T cell infiltration” or simply T cell numbers ”Results page 14 second paragraph – “foregoing” does not make sense.

A12: We have made these wording changes and highlighted them per reviewer’s suggestion: “modality” has been corrected to “mortality”; “T cell frequency” has been changed to “T cell infiltration”; “foregoing” in *Results* page 16 (originally P14) has been changed to “above”.

Q13: *Figure 7 - what is IRS*

A13: IRS is a Semiquantitative immunoreactive score for IHC analysis, which means PP (score of percentage of positive cells) ×SI (score of staining intensity). The detailed information is presented in *Methods*.

Q14: *Discussion p18 – “annulling” is not the correct word and needs to be changed; UB55 should be UBR5; “Reconstitution” is not correct word, see below. Methods p21 – “reconstituted” (here and throughout the manuscript) is not the correct word. Suggest using something like “re-expression of Ubr5” in Ubr5-/- cells and “over-expression of Ubr5” in parental cell.*

A14: “Annulling” has been changed to “depletion of”; “reconstitution” has been changed to “reintroduction” throughout the manuscript.

Q15: *Page 22 Methods – “since day” should be “from day” Methods for cell viability and sulforodamine assays appear to be missing. Data and materials availability - this statement needs clarification as it does not make sense. Access to the data needs to comply with the journal requirements.*

A15: (1) “since day” has been changed to “from day”.

(2) The method for cell viability and the sulforhodamine B (SRB) assay has been added in *Methods* and highlighted.

(3) The statement in *Data and materials availability* has been modified to comply with the journal’s requirements.

Reviewer #3:

Q1: *Author considered β -catenin as epithelial marker and claimed that loss of β -catenin in UBR5 deficient cells demonstrated epithelial to mesenchymal transition. On contrary, many other studies suggest activation of Wnt/ β -catenin signalling induces EMT - (1) it down-regulates E-cadherin expression via the transcription factors Twist and Slug, (2) it up-regulates adhesion molecules that favor cell motility, such as N-cadherin and L1, and (3) it induces proteases and other EMT promoters. Wnt signaling can therefore induce a cadherin switch and weaken cell–cell adhesion (PMID: 20182623, 31101875, and 30271576).*

Additionally, the author claimed that “It was noteworthy that knocking down p53 expression in ID8/Ubr5-/- cells restored β -catenin to the control level (Fig.5g)”, which is in corroboration with the previous publication, which also claimed that loss of p53 restores the transcriptional activity of β -catenin which further induce EMT (PMID: 24023784). What is the status of EMT markers in ID8/Ubr5-/- after the restoration of β -catenin (if the author claims β -catenin as an epithelial marker) by p53 knockdown?

A1: (1) We compared E-cadherin/N-cadherin expression between ID8/GFP and ID8/*Ubr5*^{-/-} cells and no significant differences were observed (originally Supplementary Fig 6d), suggesting that *Ubr5* deficiency didn’t induce cadherin switch and subsequent EMT.

(2) Per reviewer's request, we analyzed the expression of E-cadherin, cytokeratin 18, and Snail in ID8 cells after *Tp53* expression knocking down. In contrast to the restoration of β -catenin expression after *Tp53* silencing, E-cadherin expression decreased by *p53* knockdown, the levels of cytokeratin 18 and Snail were not altered. Other mesenchymal markers (including N-cadherin, Vimentin, and fibronectin) were hardly detectable. We added this data in Supplementary Fig. 7d.

Q2. *According to results in figure 1, the author says URB5-deficient cells show more EMT and undergo more metastasis in lung till day 30. But, because of impaired MET, URB5-deficient cells do not form macro-metastasis in the lung. The finding should be confirmed by analysis of more EMT markers (like E-cadherin, Cytokeratin 18, ZO-1, Fibronectin 1, Vimentin, N-cadherin, Snail, Zeb1) on protein level in vitro results. Additionally, the author needs to perform IFC analysis (EMT markers+ID8+UBR5) in the lungs to claim EMT at day 30 and impaired MET at day 60 tumors, and to show ID8/Ubr5^{-/-} cells are survived in lung microenvironment after 60 days in circulation. Because in the discussion, the author claimed that "UBR5 has been shown to suppress death receptor-induced apoptosis and downregulate proapoptotic MOAP-1 in human OC cells. However, we observed that genes involved in apoptotic pathways were only slightly altered by Ubr5 deficiency at the mRNA level (data not shown)."*

A2: (1) The protein expression of E-cadherin, N-cadherin and ZEB1 are shown in Fig 1f, and supplementary Fig. 6d respectively. Per the reviewer's request, we observed that epithelial marker Cytokeratin 18 was downregulated after *Ubr5* silencing in ID8, while the protein level of the mesenchymal marker vimentin was increased in ID8/*Ubr5*^{-/-}. However, vimentin expression was very low in control ID8, and the other two mesenchymal markers ZO-1 and Fibronectin 1 were undetectable in ID8. We incorporated these results in Fig 1f.

(2) Per reviewer's request, we performed IF staining in lung sections at day 30 post *i.v.* injection, which revealed β -catenin expression in ID8/GFP cells, but not in ID8/*Ubr5*^{-/-} cells, while higher levels of vimentin were observed in ID8/*Ubr5*^{-/-} tumors at day 60 post *i.v.* injection. Consistent with the FACS data showing that there were more ID8/*Ubr5*^{-/-} tumor cells in the lungs compared with control tumor cells (Fig 1c), IF staining results also show that there were at least comparable numbers of tumor cells in the lungs of ID8/*Ubr5*^{-/-} tumor bearing mice at day 60 post *i.v.* injection, compared to the control group. These data are included in Supplementary Fig2i, j. In addition, ID8/*Ubr5*^{-/-} tumor bearing mice did not exhibit enhanced TUNEL-positive signals in the lungs with intravenous injection, compared to ID8/GFP tumor bearing mice (Fig 5e).

Q3: *In the whole manuscript, the β -catenin level was checked by using whole cell extract, which may indicate the cytoplasmic concentration of β -catenin (transcriptionally inactive). At least critical experiments in the manuscript (fig.1f, fig. 5g, fig 6c) need to show the transcriptional activity of β -catenin either by WB of β -catenin in nuclear and cytoplasmic fraction or TCF/LEF activity assay along with protein expression of Wnt target genes like CCND1, MYC; etc.*

A3: Per the reviewer's requests, we performed western blot to show β -catenin protein levels in nuclear and cytoplasmic fractions. Results have been incorporated in Fig. 5g and Fig. 6c, respectively. We do not present these results in Fig1f due to limited space.

Q4: *What is the nodule count at day 30 in figure 1e?*

A4: We have added the count in Fig 1d.

Q5. *The heat map in supp. fig 5a need to show the name of genes are in G1 and G2?*

A5: We have now presented detailed genes in G1/G2 of Supplementary Fig6a (formerly Supplementary Fig5a) in Source data file due to the large size of the data. We have also performed Ingenuity Pathway Analysis (IPA) of the G1- and G2-contained genes. The result of the analysis has been appended into Fig S6a.

Q6. *The author demonstrated that “TAMs from Ubr5^{-/-} tumor-bearing mice exhibited higher levels of M1 macrophage markers (e.g. il12a, Ccr2) and lower levels of M2 markers (e.g. Cx3cr1, il10) (Fig.4b and Supplementary Fig.5b).” Change in macrophage phenotype in the tumor microenvironment does affect the phenotype of T cells, so what is the explanation behind not observing the significant shift in the infiltration or phenotype of any other immune cells?*

A6: Although there were little differences between ID8/GFP and ID8/*Ubr5*^{-/-} tumors in the proportion of immune subsets in the TEM except TAMs, the absolute number of all infiltrating immune subsets were decreased in ID8/*Ubr5*^{-/-} tumors (Supplementary Fig. 4i). However, we found that decreased TAMs, not T cells, were responsible for the abrogated ID8/*Ubr5*^{-/-} tumor growth, a conclusion based on using RAG2^{-/-}, CD4⁺ T, and CD8⁺ T-deficient mice (Fig 2d), and adoptive transfer of TAMs (Fig 3).

Q7. *According to figure 6a, Ccl2, Vegf, Il6, Csf-1, Cxcl1, Genes in Wnt signaling were significantly down-regulated in ID8/Ubr5^{-/-} bearing mice (Fig.6a). Therefore, the author reintroduces CCL2, M-CSF, and β-catenin in ID8/Ubr5^{-/-} cells. But according to figure 6c, the reintroduction of CCL2, M-CSF, and β-catenin do not restore protein expression of UBR5 in ID8/Ubr5^{-/-} cells (pink block). In contrast, all UBR5-related functional effects (Fig.6 d-f and Supplementary Fig.6 e-i) were restored. Is it means UBR5-related functional results are independent of the protein level of UBR5?*

A7: The reintroduction of CCL2, M-CSF and β-catenin in ID8/*Ubr5*^{-/-} cells only partially restored the *in vitro* cellular function and *in vivo* growth of ID8/*Ubr5*^{-/-} tumors, while reintroduction of hUBR5 in ID8/*Ubr5*^{-/-} could fully restore tumor growth to the control level. This partial restoration of the cellular function and tumor growth in ID8/*Ubr5*^{-/-} cells by CCL2/ M-CSF/β-catenin reexpression suggests that these three factors contribute significantly but not completely to UBR5-mediated OC progression. In other words, additional factors regulated by UBR5 may also be involved, which is currently under investigation.

Q8. *Figure 6 showed the combined effect of the reintroduction of CCL2, M-CSF, and β-catenin in ID8/Ubr5^{-/-} cells. What are the results, if CCL2, M-CSF, and β-catenin reintroduce separately in ID8/Ubr5^{-/-} cells?*

A8: We compared tumor growth of ID8/*Ubr5*^{-/-}+β-catenin, ID8/*Ubr5*^{-/-}+CCL2/MCSF with ID8/*Ubr5*^{-/-}+β-catenin+CCL2/MCSF. Separate reintroduction of β-catenin, or CCL2/MSCF could not significantly restore ID8/*Ubr5*^{-/-} tumor growth (data not shown).

Q9. *In the discussion, author claimed that “Given that UBR5 didn’t modulate their mRNA stability (Supplementary Fig.6c), UBR5 may stabilize transcription factors (TFs) that control Ccl2/Csf1 gene transactivation or act as coactivator of these TFs”. In the present study, the author also showed modulation of β-catenin (which is also a TF) in UBR5-deficient cells. It is important to check the role of UBR5-dependent β-catenin regulation in Ccl2/Csf1 gene transactivation because β-catenin is known to play a role in transactivation of CCL2 (PMID: 16003740).*

A9: Per the reviewer’s requests, we reintroduced β-catenin expression in both ID8/GFP and ID8/*Ubr5*^{-/-} by transient transfection, which increased CCL2 expression (Supplementary Fig. 8i), suggesting that β-catenin may well play a role in transactivation of *Ccl2*. In addition, knocking down *Tp53* in ID8 cells not only increased β-catenin protein expression, but also enhanced CCL2 production (Supplementary Fig.8j),

indicating an UBR5 dependent p53- β -catenin-CCL2 axis in ID8.

Q10. The previous publication from the author (E3 Ubiquitin Ligase UBR5 Drives the Growth and Metastasis of Triple-Negative Breast Cancer) showed similar findings in breast cancer. Why the author did not explore UBR5 dependent molecular and functional role in the modulation of TAMs (like fig 2), cytokines, chemokines, and β -catenin (like fig. 5 and 6), Chemotherapeutic and Immuno-Therapeutics therapeutic advantages (like fig. 7 and 8) in breast cancer. What is the rationale behind performing follow up study in ovarian cancer and not in breast cancer? Are the molecular events demonstrated in the present study are only observed in ovarian cancer?

A10: Yes, indeed, we found that the key UBR5-mediated mechanisms involved in the two tumor models were different. In the 4T1 mammary tumor model, the major mechanism whereby UBR5-deficiency caused impairment in tumor growth involved diminished CD8⁺ T cells in a causative manner (Song et al. Oncoimmunology 2020, PMID: 32363114), whereas in the ID8 ovarian cancer model in the present work, it was principally impaired TAMs that underlie abrogated tumor growth affected by UBR5-deficiency. What causes are behind the different mechanisms in the two models are currently under investigation.

All changes are highlighted in the revised manuscript.

REVIEWERS' COMMENTS

Reviewer #1 (Remarks to the Author):

I am pleased to see the authors had taken a tremendous effort to improve the quality and presentation of this manuscript. Most of my recommendation was adequately addressed, particularly adding the requested locally formed ID8 models with intraovarian injection.

However, I don't think my comments on requesting the primary OC mets at liver and mesentery in addition to lung were not valid. ID8 is a murine high-grade serous ovarian cancer cells, while can spread via multiple routes including hematogenous and lymphatic metastasis. The most common distal sites for HGSC are reportedly as lymph nodes, liver and lung. Both the intravenous ID8/Ubr5^{-/-} vs wt models and intraovarian models, I would like authors to present the additional mets counts on these sites, which can direct support their conclusion of the role of UBR5 in tumor metastasis.

Reviewer #2 (Remarks to the Author):

The authors have addressed my comments. I have no more comments.

Reviewer #3 (Remarks to the Author):

Authors addressed all my comments to satisfaction. Thus, I recommend this manuscript for publication.

** See Nature Research's author and referees' website at www.nature.com/authors for information about policies, services and author benefits

Attachment: NCOMMS-20-04534A Extended comments.docx - 20th October 20 06:27:37

Attachment: NCOMMS-20-04534A_Reporting summary_completed.pdf Source File pdf - 20th October 20 06:27:45

Print Email

Point to Point Response

We are very grateful of all three reviewers' unanimous recognition of and satisfaction with the improvement of the manuscript by the revision. Since Reviewers #2 and 3 have no more concerns, we will hereby address Reviewer#1's remaining issue as follows.

Reviewer #1's Remarks to the Author):

"I am pleased to see the authors had taken a tremendous effort to improve the quality and presentation of this manuscript. Most of my recommendation was adequately addressed, particularly adding the requested locally formed ID8 models with intraovarian injection. However, I don't think my comments on requesting the primary OC mets at liver and mesentery in addition to lung were not valid. ID8 is a murine high-grade serous ovarian cancer cells, while can spread via multiple routes including hematogenous and lymphatic metastasis. The most common distal sites for HGSC are reportedly as lymph nodes, liver and lung. Both the intravenous ID8/Ubr5^{-/-} vs wt models and intraovarian models, I would like authors to present the additional mets counts on these sites, which can direct support their conclusion of the role of UBR5 in tumor metastasis."

Response: We have now added new data to Fig 1d, which clearly shows that liver metastasis, like in the lungs, was also severely compromised in UBR5-deficient ID8 tumor injected i.v. into recipient mice. The data is consistent with a strong role of UBR5 in directly influencing tumor metastasis. We will conduct a much larger and comprehensive analysis of this nature to test a PROTAC inhibitor of UBR5 that has been developed at our institution in close collaboration with our pharmaceutical partners.